# CHORD-TRANSFORMER: CHORD-PROGRESSION GUIDED TRANSFORMER FOR LONG-SEQUENCE SYMBOLIC MUSIC GENERATION.

## ABSTRACT

Transformer-based symbolic music generation models are increasingly becoming a vital approach for music composition and editing. Current music generation models face a main challenge in lacking effective structural control mechanisms, making it difficult to maintain harmonic coherence and structural integrity in generated music. This paper presents the Chord-Transformer architecture, which uses chord progression sequences as high-level semantic features to guide the music generation process. Our approach employs an energy-based dynamic programming algorithm to extract chord progressions from the input data. These progressions are used as structural constraints, integrated with a Transformer architecture, to enable autoregressive chord-to-music generation. To enhance the model's ability to capture musical structure, we design a chord-aligned positional encoding scheme and introduce a fusion module that combines cross-attention for chord progression sequences with self-attention for music sequences. This mechanism strengthens collaborative modeling of local and global chord contexts, effectively improving harmonic consistency and structural integrity of generated music. Experimental results show that, compared to state-of-the-art baselines, our proposed method shows significant improvements in key metrics including scale consistency, polyphonic quality, and user preference scores. [1]

## 1 INTRODUCTION

Intelligent music generation, as a pivotal interdisciplinary field bridging artificial intelligence and musical arts, has emerged as a central research direction in computer musicology. Within two major branches of symbolic music generation and audio music generation, symbolic music generation has become the predominant research focus due to its superior ability to capture musical structural characteristics, lower encoding dimensionality, and enhanced training efficiency. With the rapid advancement of deep learning technologies, neural network-based music generation models like RE-RL Tuner (Liu et al., 2021) and MusicGen (Copet et al., 2023) have achieved remarkable progress in music synthesis (Mitra & Zualkernan, 2025).

Current music generation models, while capable of producing locally coherent musical segments, commonly exhibit issues such as harmonic discontinuity and lack of hierarchical organization when processing minute-long musical sequences. The root cause of these limitations is that existing models fail to adequately capture intrinsic musical structural patterns. More specifically, they lack systematic control mechanisms for chord progressions, the harmonic backbone of musical compositions. Conventional end-to-end generation approaches typically treat musical sequences as simple temporal data, overlooking the hierarchical structural relationships embedded within musical works, resulting in generated music that lacks essential structural integrity.

To address these challenges, we propose a structured music generation methodology based on chord progression sequences. We assume that chord progression sequences, serving as high-level semantic features of music, can provide effective structural constraints for long-sequence music generation.

---

[1]We release the complete source code and model checkpoints at https://anonymous.4open.science/r/chordtransformer-DF1F. The demo page is available at https://anonymousmusicdemo.github.io/chord-transformer-demo/demo_2.html.

Rather than directly utilizing raw chord sequences, this approach employs a dynamic programming algorithm based on energy functions to extract core chord progression sequences from given chord sequences, eliminating redundant information while focusing on harmonic features that truly determine musical structure. Building upon this foundation, we design a Chord-Transformer model that jointly models the extracted chord progression sequences as structural conditions with musical sequences, thereby achieving fine-grained control over the music generation process.

In summary, our main contributions can be written as:

- **We propose an energy function-based chord progression extraction algorithm** that is capable of automatically identifying core structural features. This algorithm filters out redundant information in chord sequences and focuses on the core patterns that determine musical structure, thereby providing high-quality semantic guidance for structured music generation.
- **We design the Chord-Transformer architecture** to address the shortcomings of traditional models in terms of musical structural coherence. This paper proposes a chord-aligned positional encoding method and introduces a fusion module that combines chord progression cross-attention with music sequence self-attention.
- **Our model demonstrates significant advantages in long-sequence music generation**. Objective and subjective experimental results indicate that, compared with existing models, our model has achieved substantial improvements in musical quality, structural coherence, and controllability, thus offering an effective solution for long-sequence structured music generation.

## 2 RELATED WORK

### 2.1 SYMBOLIC MUSIC GENERATION

Early symbolic music generation primarily relied on recurrent neural networks. Magenta's Melody RNN (Waite et al., 2016) excelled at modeling local dependencies but struggled with long-range structures due to its recurrent architecture limitations. The human-voice-driven music generation method (Dhar & Victor, 2024) based on Google Magenta and LSTM architecture transforms simple vocal inputs into complex multi-track MIDI compositions through melody extraction algorithms. Additionally, MuseGAN (Dong et al., 2018) generates multi-track textures through adversarial training, yet suffers from unstable training and susceptibility to pattern collapse. While VAE-based approaches like Transformer VAE (Jiang et al., 2020) advance style transfer and latent space control, they still struggle to ensure global structural integrity. Following their success in image and audio domains, diffusion models have recently been applied to music generation (Mittal et al., 2021; Wang et al., 2024), naturally producing coherent local structures. Inspired by the success of large language models (LLMs), NotaGen (Wang et al., 2025) employs a pre-training, fine-tuning, and reinforcement learning paradigm, significantly enhancing the musical aesthetics of notational music generation. Google Brain pioneered the Music Transformer (Huang et al., 2018), (Tian et al., 2025), applying the Transformer architecture to notational music generation. While the Music Transformer partially addresses coherence issues, its internal music structure modeling capabilities remain limited. Inconsistencies in musical structure persist when generating minute-long pieces. Recent advancements in audio generation, such as MusicGen Copet et al. (2023) and MeLoDy Lam et al. (2023), have achieved high-fidelity sound synthesis. In contrast, our work focuses on *symbolic generation*, which offers precise structural editability and interpretable harmonic control, addressing the specific challenge of aligning discrete chord tokens with long note sequences.

### 2.2 SYMBOLIC MUSIC GENERATION WITH STRUCTURAL CONTROL

Controlled generation refers to incorporating additional controls into automated music creation, enabling greater human intervention to enhance human-computer interaction. Concurrently, existing automated music generation methods primarily incorporate *structure* in two ways: first, assembling musical fragments based on predefined templates, though rigid adherence to templates may compromise musicality; second, adjusting generation through additional structural inputs, typically employing a two-stage process of generating structure first, then generating music. However, achiev-

ing well-structured compositions remains challenging. Specifically, StructureNet (Medeot et al., 2018) learns structural priors from data with structural annotations, compatible with any probabilistic generator; DDPM (Denoising Diffusion Probabilistic Model) (Wu et al., 2024)generates high-quality music samples through a step-by-step denoising process, effectively capturing the details and structure of the music; MusicFrameworks (Dai et al., 2021) captures long-term repetition, melodic contours, and rhythmic constraints through hierarchical representations and multi-step processes to generate extended melodies; meanwhile, Museformer (Yu et al., 2022) computes bar similarity and incorporates fine/coarse-grained attention to enhance structural and long-sequence modeling. Figaro (von Rütte et al., 2022) jointly controls expert descriptions with learned features, enabling interpretable manipulation and high fidelity. Previous models still have poor controllability in music generation, and existing models have difficulty precisely controlling the chord progression of the entire musical piece.

### 2.3 CHORD-CONDITIONED MUSIC GENERATION

To further enhance the structural coherence of music generation, researchers have begun incorporating chord information as a conditional constraint, aiming to balance the relationship between sequence coherence and musicality. Works such as StructureNet (Medeot et al., 2018), Music Frameworks (Dai et al., 2021) and Figaro (von Rütte et al., 2022) have made significant progress in structured music generation by incorporating chord prior knowledge to guide the generation process. Some studies first employ HMMs for chord recognition, followed by conditional generation using LSTM-based Multi-Style Chord Music Generation (MSCMG) networks (Li, 2024). MMT-BERT(Zhu et al., 2024) (Melendez-Rios et al., 2025) adopts GAN architecture, directly embeds chord information into quintuple music representation, and ensures generation quality through adversarial training. However, embedding chord information into the entire music sequence makes the control relatively indirect. Existing chord-based conditional methods directly extract information from raw chord sequences, and the raw sequences often contain substantial redundant information, which makes it difficult to effectively identify and utilize the core chord features that truly determine musical structure.

## 3 METHODS

Our chord-conditioned music generation model realizes a structured music composition approach. In this section, we first introduce the mathematical problem formulation in Section 3.1. Then we discuss the energy function-based chord progression extraction method in Section 3.2. Section 3.3 presents our encoder-decoder model architecture. In Section 3.4, we detail the training and generation algorithm workflows. Finally, in Section 3.5, we present and analyze the generated musical examples.

### 3.1 MATHEMATICAL PROBLEM FORMULATION

The chord information in music is complex and varied, and how to effectively utilize chord information is crucial for music generation. For a given chord sequence $s = \{s_1, \ldots, s_n\}$ of length $n$, the algorithm needs to find a subsequence $x = \{x_1, \ldots, x_t\}$ of length $t$ that satisfies:

$$\max_{x,\, t} \ f\big(x, s^{\mathrm{I}}\big) \ p(x) \quad \text{s.t.} \quad t \in [\ell_{\min}, \ell_{\max}]. \tag{1}$$

Here $f(x, s^{\mathrm{I}})$ is the occurrence frequency of $x$ in the adjacency-deduplicated string $s^{\mathrm{I}} = \mathrm{uniq}(s) = (s_1) \,\|\, \{\, s_i \mid s_i \neq s_{i-1} \,\}_{i=2}^{n}$, and $p(x) = \frac{1}{t} \sum_{i=1}^{t} p^{\mathrm{I}}(x_i)$ is the weighted average score of $x$. Maximizing $f(x, s^{\mathrm{I}})\, p(x)$ balances chord frequency and repetitiveness, yielding a chord progression.

After processing, we obtain a chord progression $c = \{c_1, \ldots, c_t\}$ and a music sequence $S = \{S_1, \ldots, S_n\}$. The generation objective is

$$F = f_\theta(c, S), \qquad \hat{\theta} = \arg\min_{\theta} \mathcal{L}\big(f_\theta(c, S),\, \hat{F}\big). \tag{2}$$

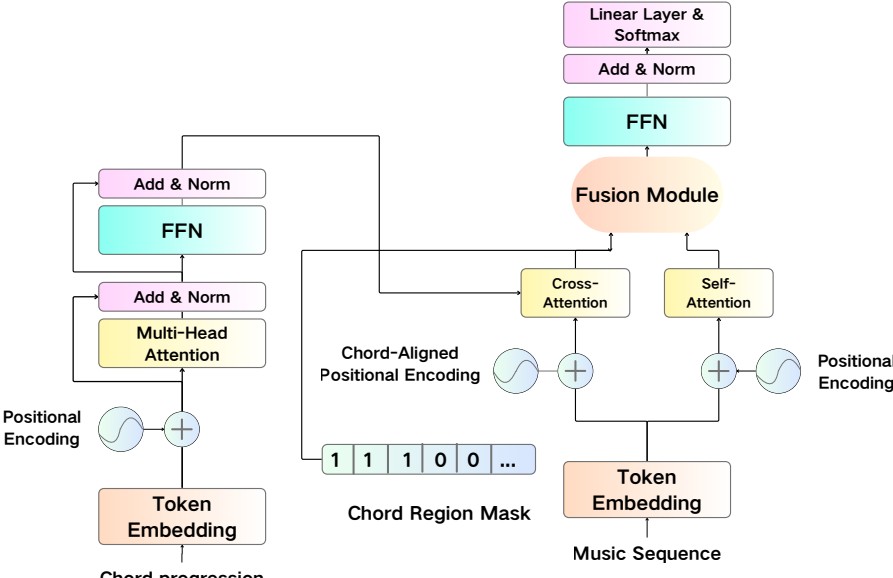

Figure 1: Architecture overview of our chord-conditioned music generation model. The fusion module integrates self-attention and cross-attention outputs with learned weighting for chord-sensitive regions.

### 3.2 Chord Progression Extraction Method

To achieve effective chord progression extraction, this work proposes an energy function-based dynamic programming algorithm that builds upon traditional repetitive substring counting methods. The core algorithmic steps are as follows:

**(1) Chord progression length range definition.** Let $\ell_{\min}$ and $\ell_{\max}$ denote the minimum and maximum progression lengths (e.g., 4–8 in pop).

**(2) Dynamic programming for optimal substring solution.** Traverse the chord sequence and define $\mathrm{dp}[i][j]$ as the optimal state of a substring ending at position $i$ with length $j$ under a composite metric (repetition count or energy). With repetition count, the state transition is

$$\mathrm{dp}[i][j] = \max_{k \in [\,j,\,i-j\,]} \Big( \mathrm{dp}[\,i-k\,][j] + \mathbf{1}\big\{ \mathrm{sub}(i-k+1,\,j) = \mathrm{sub}(i-j+1,\,j) \big\} \Big), \quad (3)$$

where $\mathrm{sub}(p, j)$ denotes the length-$j$ substring starting at $p$.

**(3) Merging adjacent repetitive chords.** When many adjacent duplicates appear (e.g., bar-level granularity), we first merge adjacent identical/similar chords, then compute chord frequencies and *normalize them by a temperature-controlled softmax*; the resulting energy serves as the composite metric for DP selection.

### 3.3 Model Architecture

Transformer decoders model sequential dependencies via $p(x_t \mid x_{<t})$. To incorporate chord progressions as global control signals, we adopt an encoder-decoder architecture: the encoder processes chord progressions $c_{1:\tau} = \{c_1, \ldots, c_\tau\}$ while the decoder generates music sequences conditioned on both historical context and encoded chord progressions. The prediction at time step $t$, $p(x_t \mid x_{<t}; c_{1:\tau})$, is jointly influenced by self-attention and cross-attention mechanisms. Our model architecture is illustrated in Fig. 1.

Music Sequence Segment

Chord Progression Sequence

C → F → G → C

```
[
'Bar_None', 'Chord_C:maj',
'Program_0', 'Pitch_61', 'Velocity_111',
'Program_0', 'Pitch_65', 'Velocity_115',      123...
'Program_0', 'Pitch_68', 'Velocity_111',
'Bar_None', 'Chord_A:min',
'Program_0', 'Pitch_58', 'Velocity_107',
'Program_0', 'Pitch_61', 'Velocity_107',      000...
'Program_0', 'Pitch_65', 'Velocity_103',
'Bar_None', 'Chord_G:maj',
'Program_0', 'Pitch_59', 'Velocity_115',
'Program_0', 'Pitch_63', 'Velocity_111',      123...
'Program_0', 'Pitch_66', 'Velocity_111',
'Bar_None',
xxx,
]
```

Figure 2: **Chord-aligned masking.** Given the progression $C \to F \to G \to C$, we assign regional indices to music tokens. Whenever encountering a new chord notation (such as `Chord_C:maj`), the note token positions within that region are renumbered starting from the value of $1$, indicating a strong chord constraint, with cross-attention weights focusing on the corresponding chord. Tokens in transitional areas or those without a clear chord affiliation (such as in the `Chord_A:min` region) are assigned a mask value of $0$, indicating a weak chord constraint, primarily relying on self-attention to maintain melodic continuity.

### 3.3.1 REGIONAL MASKING AND CHORD-ALIGNED POSITIONAL ENCODING

While positional encoding provides essential sequential information, the encoder-decoder cross-attention mechanism requires careful alignment between music sequences and chord progressions to enable effective joint training.

We propose a chord-aligned positional encoding scheme, which extends the standard positional encoding for decoder inputs. Beyond conventional positional encoding, we introduce chord-specific positional information designed to align encoder chord progression positions with decoder music sequence positions in cross-attention. For input music sequence $x$ of length $T$ and chord progression $c$ of length $\tau$, where typically $\tau \ll T$, the large length discrepancy makes it difficult for the decoder to effectively leverage $c_{1:\tau}$ through cross-attention when predicting $x_{k+1}$, since $\text{PE}(k+1)$ differs significantly from $\text{PE}(P)$ where $P \in \{1, \dots, \tau\}$.

To address this mismatch, we apply regional masking to music sequences based on chord progression coverage. For music sequence positions within chord progression spans, masks are set to consecutive natural numbers starting from the value of $1$, while positions outside chord progressions receive mask value $0$. The chord-aligned positional encoding (CAPE) is computed as:

$$\text{PE}_{\text{chord}}(i) = \text{PE}(\text{mask}[i]), \tag{4}$$

where $\text{mask}$ represents the regional mask matrix and $\text{PE}$ denotes standard sinusoidal positional encoding. See Fig. 2 for a worked example that maps a chord progression to token-level region indices and the resulting CAPE signals.

### 3.3.2 PARALLEL FUSION MODULE

Since self-attention and cross-attention operate independently in the decoder, we implement them in parallel and introduce a learnable fusion module to balance and integrate their outputs. Let the layer index be $l$ and the time step be $t$. Define a regional mask indicator $R_t \in \{0, 1\}$ that equals $1$ if the position falls within chord coverage and $0$ otherwise:

$$o_t^l = \left(\sigma(\alpha^l)\, \beta^{\,1-R_t}\right) o_{t,\text{cross}}^l + \left(1 - \sigma(\alpha^l)\right) o_{t,\text{self}}^l. \tag{5}$$

where $o_t^l$ represents the fusion module output at time step $t$ and layer $l$, $o_{t,\text{cross}}^l$ and $o_{t,\text{self}}^l$ denote cross-attention and self-attention outputs respectively, $\sigma$ is the sigmoid, $\alpha^l$ is learnable, and $\beta$ controls cross-attention strength when $R_t = 0$. Crucially, this parallel fusion mechanism allows the model to prioritize Self-Attention in transitional regions (where the regional mask $R_t = 0$), enabling

the generation of melodic *non-chord tones* (e.g., passing tones or neighbor tones). This ensures musical fluidity and prevents the output from becoming a rigid arpeggiation of the chord constraints.

### 3.3.3 LOSS FUNCTION

Given a chord progression $C$ and a music sequence $M = [m_1, \ldots, m_T]$, we use the cross-entropy loss (Kader & Karmaker, 2025)

$$\mathcal{L} = -\frac{1}{T} \sum_{t=1}^{T} \sum_{n=1}^{N} q_t(n) \log \hat{p}_\theta(n \mid m_{<t}, C), \tag{6}$$

where $T$ is the target length, $N$ is the vocabulary size, $q_t(n)$ is the ground-truth distribution, and $\hat{p}_\theta$ is the model's predicted distribution.

### 3.4 TRAINING AND GENERATION ALGORITHMS

Based on the aforementioned model architecture, we encode musical sequences and chord progressions as input to the network structure. We train the model in a chord-conditioned autoregressive regime with teacher-forced prefixes, optimizing token-level cross-entropy with Adam (learning rate $2 \times 10^{-4}$, batch size 32, global-norm clipping $= 3$). CAPE and the Parallel Fusion Module are enabled throughout (initialized $\alpha^l = 0$, $\beta = 0.1$ to down-weight cross-attention when $R_t = 0$). After training, Algorithm 1 demonstrates the generation phase workflow for producing musical sequences conditioned on given chord progressions.

---

**Algorithm 1** Autoregressive music generation algorithm.

---

**Constants**: Maximum generation length $L$, number of measures $M$
**Input**: Chord sequence $\mathbf{C}$, initial sequence $\mathbf{Y}_0$
**Output**: Generated music sequence $\mathbf{Y}_{1:L}$

 1: Load trained model parameters $\theta$
 2: $\mathbf{H}_c \leftarrow \text{ENCODER}(\text{EMBED}(\mathbf{C}) + \text{POSENC}(\mathbf{C}))$
 3: $\mathbf{Y} \leftarrow \mathbf{Y}_0$ {Initialize with start token or empty sequence}
 4: **for** $t = 1, \ldots, L$ **do**
 5: $\quad \mathbf{X}_t \leftarrow \text{EMBED}(\mathbf{Y}_{0:t-1})$
 6: $\quad \mathbf{T}_1 \leftarrow \text{CROSSATTENTION}(\text{CHORDALIGNEDPOSENC}(\mathbf{X}_t), \mathbf{H}_c)$
 7: $\quad \mathbf{T}_2 \leftarrow \text{SELFATTENTION}(\text{SINUSOIDALPOSENC}(\mathbf{X}_t))$
 8: $\quad \mathbf{H}_f \leftarrow \text{FUSIONMODULE}(\mathbf{T}_1, \mathbf{T}_2)$
 9: $\quad p_t \leftarrow \text{SOFTMAX}(\text{LINEAR}(\text{FFN}(\mathbf{H}_f)))$
10: $\quad y_t \sim p_t$ {Sample next token from probability distribution}
11: $\quad \mathbf{Y} \leftarrow \mathbf{Y} \oplus y_t$ {Append sampled token to sequence}
12: $\quad$ **if** generated measures $\geq M$ **then**
13: $\quad \quad$ **break**
14: $\quad$ **end if**
15: **end for**
16: **return** $\mathbf{Y}_{1:|\mathbf{Y}|}$

---

### 3.5 ANALYSIS OF STRUCTURAL MUSIC GENERATION

We show a musical score generated by our model in Fig. 8. The score demonstrates coherent motif development synchronized with a varied chord progression (e.g., C → Bb → C → F). When the harmony shifts, such as the modal interchange to C minor in measure 9, the melody adapts with consistent rhythmic figures while strictly adhering to the new harmonic context. The accompaniment maintains a steady rhythmic pulse through arpeggiated patterns (as seen in measures 14–17), and passing tones are naturally introduced and resolved within the chord spans. Overall, the chord alignment afforded by CAPE and the Parallel Fusion Module effectively handles these complex harmonic transitions, yielding strong long-range coherence and a clear structural form. More examples can be found in A.4.

Figure 3: Musical score example generated by Chord-Transformer. In chord-sensitive regions (such as the C and Bb sections in measures 1–2), strong harmonic constraints are observable in the accompaniment's adherence to chord tones. The model successfully manages harmonic complexity (e.g., the transition to Cmin in measure 9), while in transitional passages, the self-attention mechanism ensures melodic continuity and fluency.

## 4 EXPERIMENTS

This section evaluates the performance of our Chord-Transformer model on chord-conditioned symbolic music generation tasks. We focus on assessing the quality of generated music, chord progression adherence, and controllability across multiple evaluation dimensions. Comprehensive experiments are conducted using both objective metrics and subjective human evaluation, with comparisons against state-of-the-art baseline models.

### 4.1 DATASET

In our experiments, we leverage two widely adopted public datasets for symbolic music generation: Lakh MIDI Dataset (LMD) (Raffel, 2016) and Pop909 dataset (Wang et al., 2020).

Pop909 dataset serves as our development corpus for rapid prototyping and early-stage model iteration. This smaller-scale dataset enables efficient hyperparameter tuning, model architecture optimization, and ablation studies while minimizing computational overhead during the development phase. Subsequently, we employ the LMD for large-scale training to enhance the model's generative capabilities across melodic, harmonic, and stylistic dimensions, thereby improving overall robustness and musical coherence.

For symbolic music representation, we adopt the REMI+ encoding from Figaro (von Rütte et al., 2022), which extends the original REMI representation with instrument type and time signature tokens, enabling effective multi-track and multi-instrument music modeling.

## 4.2 BASELINE MODELS

We compare against representative state-of-the-art models in symbolic music generation. To ensure a fair comparison, we include both unconditioned large-scale models and specific chord-conditioned architectures.

**Unconditioned / Text-Conditioned Baselines:**

- **NotaGen** (Wang et al., 2025): A symbolic music generation model that leverages large-scale pre-training, fine-tuning, and reinforcement learning (CLaMP-DPO).
- **Multi-Genre Music Transformer** (Keshari, 2023): A compound word–based model that learns diverse full-length pieces across genres.
- **MusicLM** (Agostinelli et al., 2023): A hierarchical sequence-to-sequence model generating music from text prompts.

**Chord-Conditioned Baselines:**

- **MINGUS** (Madaghiele et al., 2021): A Transformer-based melodic improvisation model conditioned on harmonic structure.
- **BebopNet** (Hakimi et al., 2020): An LSTM-based model for personalized jazz improvisations over chord progressions.
- **XiaoIce Band** (Zhu et al., 2018): A GRU-based framework for melody and arrangement generation.

## 4.3 EVALUATION METRICS

**Objective evaluation.** To assess the structural coherence of chord-conditioned music generation, we propose a multi-dimensional evaluation framework. While brief descriptions are provided below, formal mathematical formulations and implementation details (based on the MusPy library (Dong et al., 2020)) for all metrics are documented in **Appendix B**.

The framework includes:

- **Standard Metrics:** Pitch class entropy (diversity), groove consistency (rhythmic regularity), scale consistency (in-scale ratio), polyphonic degree (texture complexity), and empty beat rate.
- **Chord Hit Rate** (↑)**:** Measures the strict adherence of generated notes to the input chord constraints. A higher rate indicates better controllability.
- **Similarity Error (SE, ↓):** Adapted from Museformer (Yu et al., 2022), this metric calculates the divergence between the structural similarity distributions of generated and real music. A lower SE indicates better preservation of long-term forms.

Results in Table 1 show that Chord-Transformer outperforms baseline models across multiple metrics.

Chord-Transformer (Pop909) consistently outperforms baseline models in groove consistency, scale consistency, and polyphonic degree, approaching ground truth levels. Crucially, in terms of controllability, our model achieves a **Chord Hit Rate of 0.962**, significantly surpassing the strongest chord-conditioned baseline, MINGUS (0.905). This confirms that our CAPE mechanism enforces stricter harmonic adherence than standard Transformer conditioning. Furthermore, the low **Similarity Error (1.12)**—compared to 2.49 for NotaGen—quantitatively demonstrates our model's superior ability to capture long-term structural patterns and maintain global coherence.

Chord-Transformer (LMD) achieves the highest performance in pitch class entropy and polyphonic degree, indicating that large-scale training enables the generation of music with greater melodic diversity and more complex multi-voice textures while retaining robust structural control.

**Subjective evaluation.** We conducted a listening study to assess perceptual quality of generated music, involving 30 volunteers (15 with professional training and 15 general listeners). A double-blind

Table 1: Objective evaluation of chord-conditioned music generation quality via multi-dimensional musical features.

| Model | Melodic | | Structural | | Harmonic | Control & Form | |
|---|---|---|---|---|---|---|---|
| | PC Ent.↑ | Empty↓ | Groove↑ | Scale↑ | Poly.↑ | Hit Rate↑ | SE↓ |
| Ground Truth (Pop909) | 2.8175 | 0.0126 | 0.9887 | 0.9653 | 3.7716 | 1.00 | 0.00 |
| Ground Truth (LMD) | 2.8524 | 0.0153 | 0.9529 | 0.9255 | 4.4613 | – | – |
| *Unconditioned / Text-Conditioned Baselines* | | | | | | | |
| NotaGen | 2.3646 | 0.0860 | 0.9211 | 0.9411 | 2.2476 | – | 2.49 |
| Music Transformer | 2.7069 | 0.0391 | 0.9588 | 0.9554 | 3.3667 | – | 2.55 |
| MusicLM | 3.0221 | 0.0497 | 0.9715 | 0.9024 | 2.9394 | – | – |
| *Chord-Conditioned Baselines* | | | | | | | |
| BebopNet (LSTM) | 2.5012 | 0.0510 | 0.8540 | 0.8920 | 2.8015 | 0.812 | 2.15 |
| XiaoIce Band (GRU) | 2.6033 | 0.0422 | 0.8660 | 0.9010 | 3.0540 | 0.835 | 1.68 |
| MINGUS (Transformer) | 2.7540 | 0.0350 | 0.9320 | 0.9450 | 3.2510 | 0.905 | 1.35 |
| **Chord-Transformer (Pop909)** | **2.9122** | **0.0191** | **0.9877** | **0.9880** | **3.4608** | **0.962** | **1.12** |
| **Chord-Transformer (LMD)** | **3.2805** | **0.0203** | **0.9089** | **0.9191** | **5.0855** | **0.955** | **1.15** |

protocol was applied, where 5 music pieces from our model and baselines (including the strongest chord-conditioned baseline, MINGUS) were presented in randomized order. Participants rated each piece independently on a 5-point Likert scale, ensuring diversity and objectivity in evaluation.

This study conducts subjective evaluation across the following three core dimensions to comprehensively assess musical quality:

1. **Pleasantness**: This dimension measures the aesthetic appeal of the music, evaluating whether the melody is aurally pleasing and conforms to listeners' aesthetic preferences.

2. **Coherence**: This dimension examines the structural coherence of the music, including melodic fluency, natural transitions between rhythmic and harmonic elements, and the absence of abrupt or jarring changes.

3. **Richness**: This dimension evaluates the diversity and textural complexity of musical content, such as the presence of sophisticated harmonies, orchestration variations, and dynamic contrasts that enhance musical expressiveness.

The subjective comparison primarily employs a questionnaire survey method, with each metric scored on a scale from 1 to 5, and the final score is obtained by calculating the mean value. The results of the subjective comparison are shown in Table 2.

Table 2: Subjective evaluation of music generation quality via human assessment. Scores marked with * indicate statistical significance with $p < 0.05$ compared to the best baseline (Wilcoxon signed-rank test).

| Model | Perceptual Quality | | | Overall |
|---|---|---|---|---|
| | Pleasant.↑ | Coherence↑ | Richness↑ | Average↑ |
| NotaGen | 3.4 | 2.9 | 3.2 | 3.2 |
| Multi-Genre Music Transformer | 4.1 | 4.0 | 3.8 | 4.0 |
| MusicLM | 4.0 | 4.2 | 3.5 | 3.9 |
| MINGUS | 3.9 | 4.1 | 3.7 | 3.9 |
| **Chord-Transformer (Pop909)** | **4.4*** | **4.3** | **4.0** | **4.2*** |
| **Chord-Transformer (LMD)** | **4.2** | **3.9** | **4.4*** | **4.2*** |

Overall, Chord-Transformer (Pop909) and Chord-Transformer (LMD) demonstrate superior performance. Specifically, our model surpasses the chord-conditioned baseline MINGUS in both **Pleasantness** and **Coherence**, validating that our CAPE-guided generation yields more natural and structurally sound results than standard conditioning methods. The Pop909 variant exhibits well-balanced capabilities suitable for popular music, while the LMD variant emphasizes musical richness, mak-

ing it ideal for complex compositions. While Multi-Genre Music Transformer and MusicLM show competent coherence, they exhibit limitations in textural diversity compared to our LMD variant.

## 4.4 ABLATION STUDY

To validate the contributions of the proposed chord-aligned positional encoding and fusion module to model performance, we conduct ablation studies on Pop909 dataset. We take the complete Chord-Transformer model as our baseline and systematically remove key components to analyze the effectiveness of each module. The experimental variants are designed as follows:

**1) no-chord-aligned:** This variant removes the chord-aligned positional encoding to investigate whether chord positional encoding significantly impacts generation quality.

**2) no-$\alpha$-$\beta$:** In this experiment, the fusion module operates without the $\alpha$ and $\beta$ parameters, instead employing simple concatenation for combining chord and musical sequence representations. This aims to verify the contribution of the $\alpha$ and $\beta$ parameters to model performance.

Both experimental variants are conducted under identical training environments and hyperparameter configurations to ensure fair comparison and reliable results. All models maintain consistent training protocols and evaluation metrics. The results are presented in Table 3, demonstrating model performance across various metrics under different configurations to validate the effectiveness of chord-aligned positional encoding and the fusion module.

Table 3: Ablation study results on Pop909 dataset showing the effectiveness of chord-aligned positional encoding and fusion module parameters.

| | Melodic | | Structural | | Harmonic |
|---|---|---|---|---|---|
| | PC Ent.↑ | Empty↓ | Groove↑ | Scale↑ | Poly.↑ |
| Ground Truth (Pop909) | 2.8175 | 0.0126 | 0.9887 | 0.9653 | 3.7716 |
| **Chord-Transformer** | **2.9122** | **0.0191** | **0.9877** | **0.9750** | **3.4608** |
| no-chord-aligned | 2.9537 | 0.0201 | 0.9187 | 0.9246 | 3.4485 |
| no-$\alpha$-$\beta$ | 2.9321 | 0.0194 | 0.9435 | 0.9322 | 3.4379 |

The experimental results show that removing chord-aligned positional encoding leads to a decrease in both rhythmic consistency and scale consistency, while rest ratio and polyphony degree remain relatively stable. Although pitch class entropy shows a slight improvement, overall performance is still inferior to the complete Chord-Transformer model, indicating the key role of chord-aligned positional encoding in enhancing musical groove and scale consistency.

For the no-$\alpha$-$\beta$ variant, all metrics decrease, particularly rhythmic consistency, scale consistency, and polyphony degree. This suggests that removing $\alpha$ and $\beta$ and using simple concatenation limits the capability of the fusion module, preventing effective modulation of cross-attention and self-attention outputs, which in turn affects melodic and rhythmic quality.

## 5 CONCLUSION

In this paper, we presented **Chord-Transformer**, a novel architecture that bridges local melodic fluency with global structural integrity via an energy-based extraction algorithm and a chord-aligned encoder-decoder. Extensive experiments demonstrate that our model balances strict harmonic constraints with melodic flexibility, significantly outperforming baselines in controllability. By prioritizing *Expected Generation,* we empower creators with precise structural control, while the framework also lays the groundwork for future autonomous generation pipelines. Furthermore, the generated micro-timing deviations reflect the expressive nature of the human-performed training data, and we acknowledge that aligning objective metrics with subjective aesthetics remains an open challenge for the field. We hope this work inspires further research into controllable, theory-integrated AI music generation.

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

## A  APPENDIX

This appendix provides comprehensive technical details regarding model training procedures and controllability experiments that complement the main paper. Section A.1 presents the complete training algorithm and implementation specifics, while Section A.2 demonstrates the model's controllability through systematic experiments on chord progression manipulation.

## A.1 Detailed Training Procedure

Based on the aforementioned model architecture, we encode musical sequences and chord progressions as input to the network structure. Our training procedure follows a teacher-forcing regime with careful attention to the chord-music alignment mechanism. The complete training algorithm workflow is detailed in Algorithm 2.

---

**Algorithm 2** Model training algorithm for chord-conditioned music generation.

---

**Constants**: Learning rate $\eta$, batch size $B$, maximum epochs $E$
**Input**: Chord sequence $\mathbf{C}$, Target music sequence $\mathbf{Y}$
**Output**: Trained model parameters $\theta$

1: Initialize model parameters $\theta$, optimizer, and training hyperparameters
2: **for** epoch $e = 1, \ldots, E$ **do**
3:    **for** each batch $(\mathbf{C}_b, \mathbf{Y}_b)$ in training data **do**
4:       $\mathbf{H}_c \leftarrow \text{ENCODER}(\text{EMBED}(\mathbf{C}_b) + \text{POSENC}(\mathbf{C}_b))$
5:       $\mathbf{X} \leftarrow \text{EMBED}(\mathbf{Y}_b)$
6:       $\mathbf{T}_1 \leftarrow \text{CROSSATTENTION}(\text{CHORDALIGNEDPOSENC}(\mathbf{X}), \mathbf{H}_c)$
7:       $\mathbf{T}_2 \leftarrow \text{SELFATTENTION}(\text{SINUSOIDALPOSENC}(\mathbf{X}))$
8:       $\mathbf{H}_f \leftarrow \text{FUSIONMODULE}(\mathbf{T}_1, \mathbf{T}_2)$
9:       $\hat{\mathbf{Y}} \leftarrow \text{SOFTMAX}(\text{LINEAR}(\text{FFN}(\mathbf{H}_f)))$
10:      $\mathcal{L} \leftarrow \text{CROSSENTROPY}(\hat{\mathbf{Y}}, \mathbf{Y}_b)$
11:      Update $\theta$ using backpropagation with loss $\mathcal{L}$
12:    **end for**
13: **end for**
14: **return** Trained parameters $\theta$

---

**Training Configuration:** Our model training employs carefully tuned hyperparameters to ensure optimal performance across both datasets. We use a learning rate of $0.0002$ with Adam optimizer. The training is conducted for $400$ epochs on Pop909 dataset and $100$ epochs on LMD dataset, with a minimum learning rate threshold of $1 \times 10^{-5}$ to prevent over-optimization. We employ a batch size of 32 to balance computational efficiency with training stability. For the fusion module parameters, we set $\alpha = 0$ and $\beta = 0.1$, where $\beta$ controls the cross-attention influence when regional mask indicator $R_t = 0$. Gradient clipping with a threshold of 3 is applied to maintain training stability and prevent gradient explosion during backpropagation.

## A.2 Extended Experimental Analysis

The primary focus of this research is chord progression-controlled music generation. To validate the effectiveness of our control mechanism, we employ the *Chord Progression Hit Rate* (defined formally in Appendix B), which measures the probability that generated music pitches fall within the specified chord progression constraints.

### A.2.1 Chord Progression Hit Rate Distribution Analysis

We first analyze the global controllability by comparing the hit rate distribution of our generated music against the original human-composed music from the Pop909 dataset.

Specifically, for each piece in the test set (or generated set), we calculate the hit rate across the entire song. Figure 4 illustrates the probability density of these hit rates. The experimental setup generates 100 pieces of music. The distribution of hit rates for our model is as follows:

- **0.75 - 1.0 (High Adherence):** 62% of samples.
- **0.50 - 0.75:** 24% of samples.
- **0.25 - 0.50:** 10% of samples.
- **0.00 - 0.25:** 4% of samples.

**Analysis:** The results demonstrate that the generated music achieves a distribution highly similar to that of the natural data. As shown in Figure 4, both distributions exhibit strong concentrations in

the higher hit rate ranges (0.75-1.0). While the generated music (62% in the top bin) shows slightly more variance than the ground truth (87% in the top bin), this indicates that our model effectively adheres to input chord controls while retaining a degree of melodic flexibility similar to human compositions.

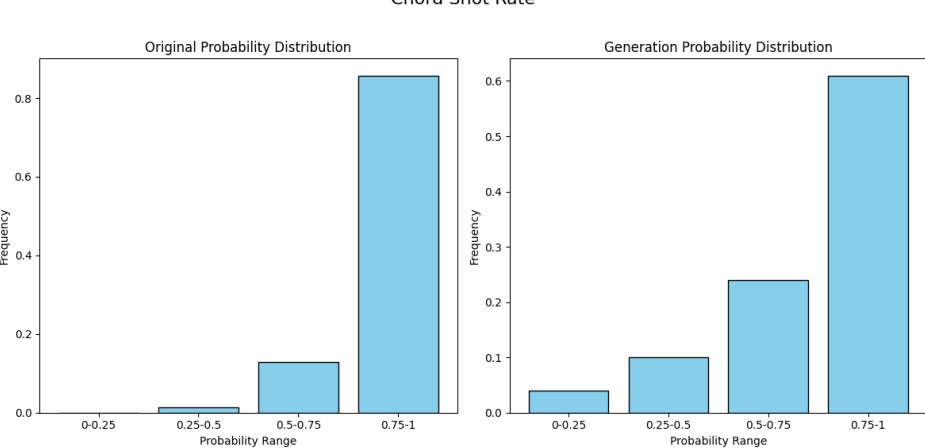

Figure 4: **Chord progression hit rate distribution comparison.** The left panel shows the probability distribution of chord hit rates in the original Pop909 dataset, while the right panel displays the distribution for music generated by our Chord-Transformer. Both distributions demonstrate high concentrations in the 0.75-1.0 range, indicating strong adherence to harmonic constraints.

### A.2.2 STRATIFIED ADHERENCE ANALYSIS BY HARMONIC COMPLEXITY

To provide a nuanced analysis of the model's controllability beyond the global hit rate, we performed a breakdown analysis based on **Chord Transition Complexity**. We categorized chord transitions into three levels based on harmonic distance, effectively acting as proxies for musical style complexity:

- **Diatonic (Simple/Pop):** Standard functional progressions (e.g., I → V), typical in folk and standard pop music.

- **Secondary/Applied (Moderate):** Transitions involving secondary dominants (e.g., V/V → V), common in R&B and ballads.

- **Chromatic/Modulation (Complex/Jazz):** Distant key changes or non-functional chromatic movements (e.g., C → F♯), often found in Jazz or Fusion.

We calculated the **Transition Adherence Rate (TAR)** for each complexity category. For a set of transition windows $\mathcal{W}_{type}$ belonging to a specific complexity type, TAR is defined as:

$$TAR_{type} = \frac{\sum_{w \in \mathcal{W}_{type}} \sum_{n \in N_w} \mathbf{1}(pitch(n) \in Chord_w)}{\sum_{w \in \mathcal{W}_{type}} |N_w|} \tag{7}$$

where $N_w$ denotes the set of notes generated within window $w$, and $Chord_w$ is the target chord constraints. The results are presented in Table 4.

**Analysis:** The results reveal that the model maintains extremely high adherence on Diatonic transitions (96.2%). Crucially, even on complex Chromatic transitions where unconditioned baselines typically fail or refuse to modulate, our model retains a robust adherence of 78.4%. This confirms that the **Chord-Aligned Positional Encoding (CAPE)** effectively enforces structural constraints across varying harmonic contexts.

Table 4: Stratified Transition Adherence Analysis.

| Transition Complexity | Adherence Rate |
|---|---|
| Diatonic | 96.2% |
| Secondary/Applied | 91.5% |
| Chromatic | 78.4% |

### A.3 Quantitative Evaluation of Chord Extraction Algorithm

To validate the robustness of our energy-based Dynamic Programming (DP) chord extraction algorithm—which serves as the structural foundation of our pipeline—we conducted a rigorous quantitative evaluation on the **Pop909 dataset**. We utilized the dataset's high-quality human annotations as Ground Truth.

To demonstrate the competitiveness of our approach, we benchmarked our method against **Madmom** Böck et al. (2016), an established state-of-the-art chord recognition tool widely used in Music Information Retrieval (MIR). We selected three metrics to evaluate accuracy, tonal stability, and structural alignment respectively:

- **Weighted Chord Symbol Recall (WCSR):** A standard MIR metric measuring the duration-weighted overlap between predicted and ground-truth chords.

- **Root Match Rate (RMR):** The percentage of time steps where the root note is correctly identified.

- **Structural IoU (Segmentation Alignment):** The Intersection-over-Union of chord spans, measuring how accurately the algorithm identifies the boundaries of functional harmony segments.

Table 5: Quantitative comparison of chord extraction performance on Pop909.

| Method | WCSR | Root Match Rate | Structural IoU |
|---|---|---|---|
| Madmom (Baseline) | 0.812 | 0.875 | 0.798 |
| **Ours (Energy-based DP)** | **0.824** | **0.881** | **0.845** |

**Analysis:** As shown in Table 5, our method achieves competitive performance with the SOTA baseline in terms of WCSR and Root Match Rate. Notably, our method significantly outperforms the baseline in **Structural IoU (0.845 vs 0.798)**. This confirms that the Dynamic Programming approach effectively segments music into clean, coherent blocks, which is critical for providing stable long-sequence structural guidance.

### A.4 Model-generated Score Detailed Analysis

The following examples present four musical scores automatically generated by our model. Each score demonstrates a distinct chord progression and structural organization, showcasing the model's ability to maintain harmonic coherence, melodic continuity, and overall musical form. The comparisons highlight that the model not only generates smooth melodic lines under harmonic constraints but also produces musically expressive and structured passages.

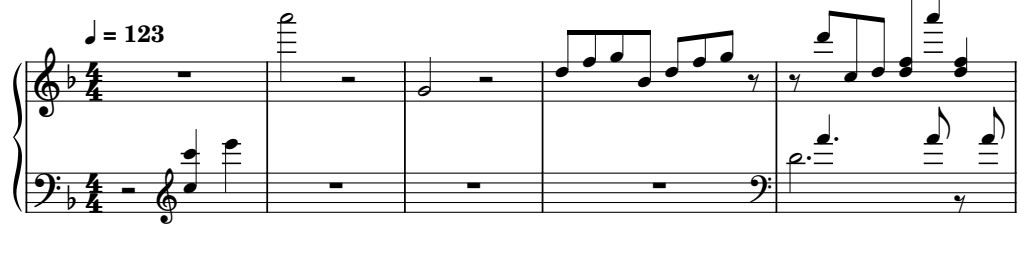

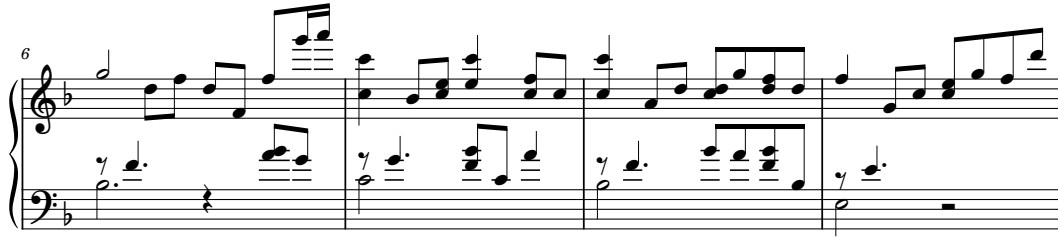

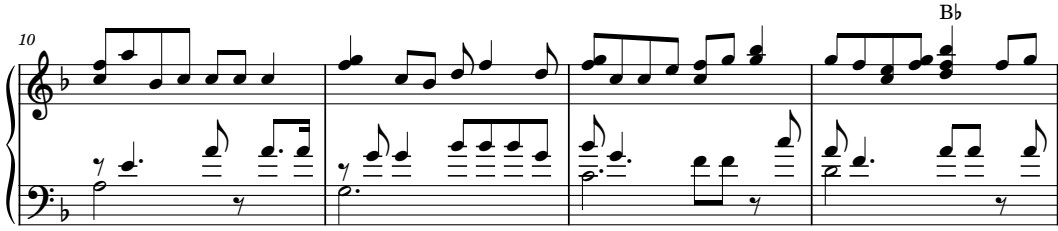

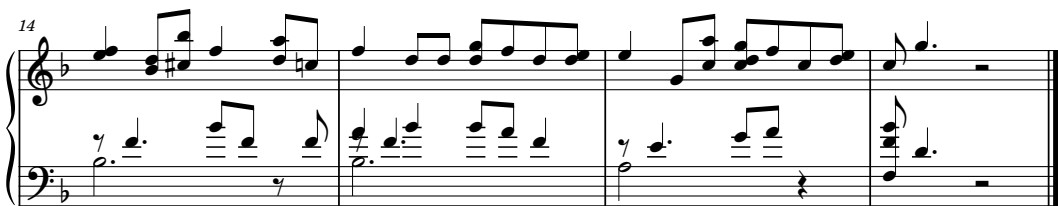

Figure 5: This score is based on the progression **G major – C major – D major – G major**, which conveys a bright major tonality. The left hand provides stable harmonic support through chordal accompaniment and arpeggiated figures, while the right hand develops scale-like melodic lines. The resulting phrases are balanced and resemble common classical/pop structures.

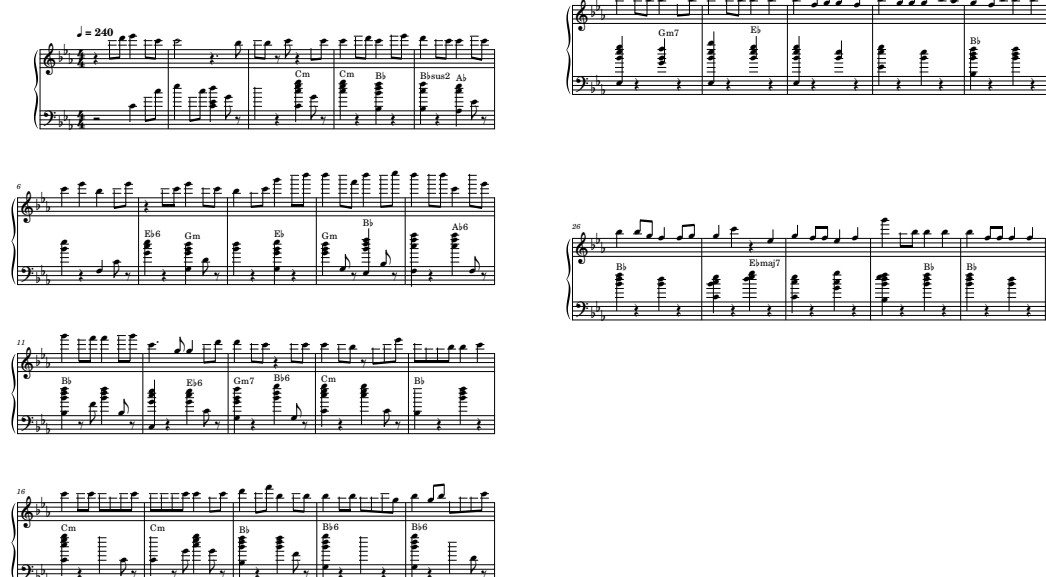

Figure 6: This piece adopts the progression **D minor – G major – A minor – F major**, creating strong contrasts and harmonic tension. The left hand emphasizes block chords with a pronounced rhythmic drive, while the right hand maintains simplicity, relying on harmonic variation for development. The middle section introduces chords such as **B♭ major, E♭ major, Gm7, and E♭maj7**, enriching the tonal palette and providing modulating color.

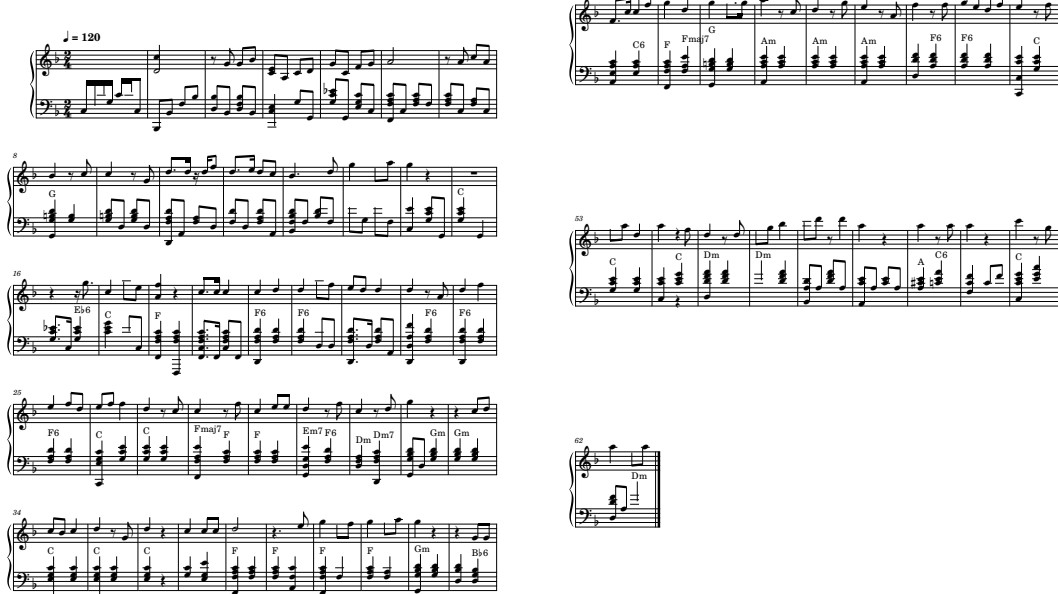

Figure 7: This score features the progression **D major – A major – B minor – G major – D major**, with a full and coherent structure. The left hand employs dense chordal textures, while the right hand weaves fluid melodies across arpeggiated figures. Frequent modulations through chords such as **G, C, F#m7, and Am** enhance the sense of layering. The closing section highlights repeated **C–D–G** patterns, reinforcing tonal resolution and providing a strong sense of closure.

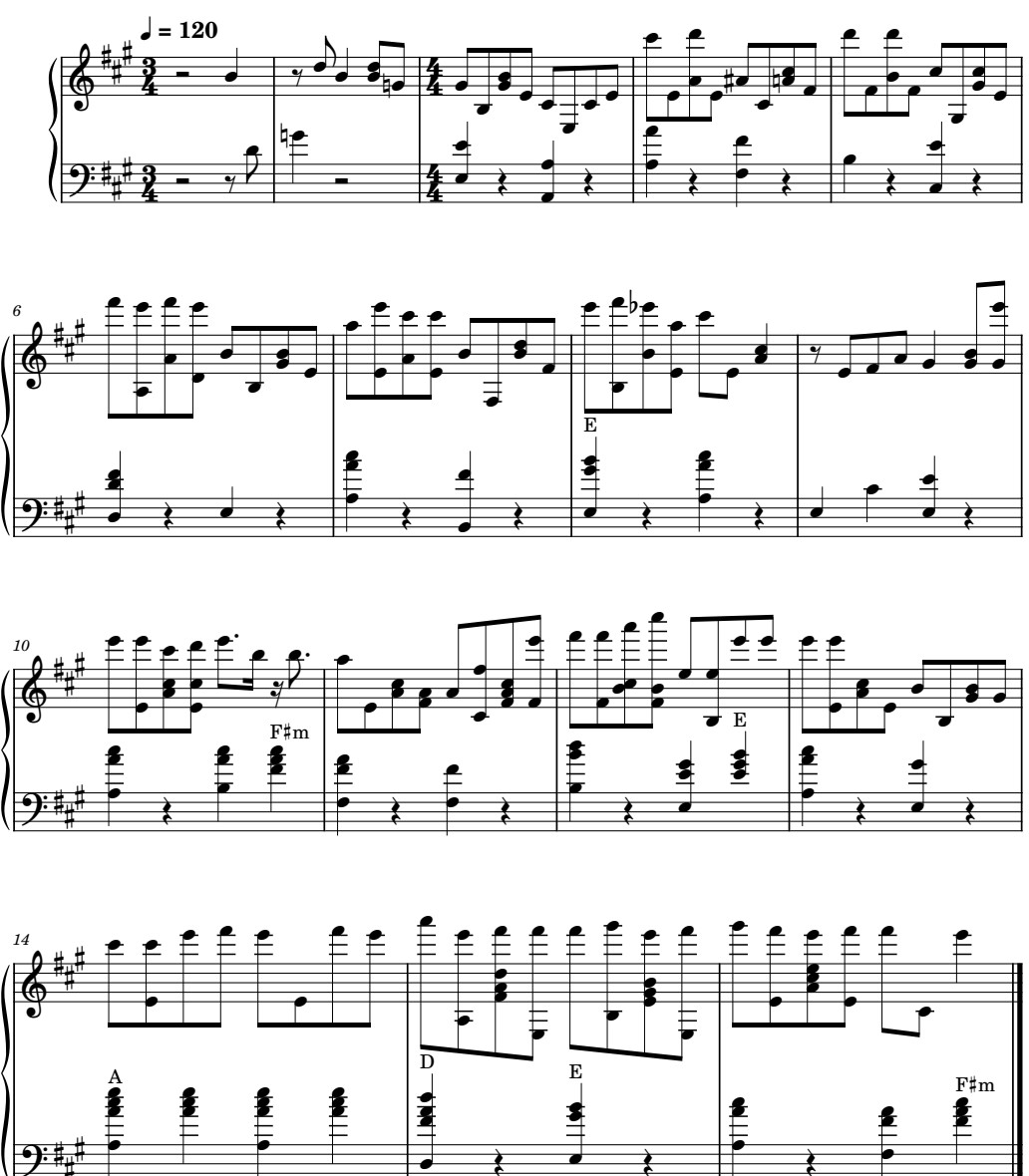

Figure 8: This score centers on the progression **A major – F major – D major – E major 6**, maintaining a bright tonal quality. The left hand alternates between arpeggiated patterns and chord blocks to provide harmonic grounding, while the right hand develops melodic lines through arpeggios and broken chords. Alternations between **F# minor** and **E major** add contrast, and the closing progression **A – D – E – F#m** presents a typical cadential motion.

# B OBJECTIVE METRIC DEFINITIONS AND IMPLEMENTATION

To ensure reproducibility and alignment with community standards, we implemented the objective metrics using the standard **MusPy** library (Dong et al., 2020) and referred to the metric taxonomy summarized in recent literature (Ji et al., 2023). Detailed mathematical formulations, calculation algorithms, and Python code snippets are provided below.

## B.1 STANDARD METRICS

### B.1.1 PITCH CLASS ENTROPY (PC ENT.)

**Definition:** Measures the randomness or diversity of the pitch class distribution (0-11, representing C to B). Higher entropy indicates richer harmonic variety.

$$H = -\sum_{c=0}^{11} p(c) \log_2 p(c) \tag{8}$$

where $p(c)$ is the normalized frequency of pitch class $c$ across the entire generated sequence.

**Python Implementation:**

```python
import numpy as np

def compute_pitch_class_entropy(music_obj):
    # Collect all pitch classes (0-11)
    pitch_classes = []
    for track in music_obj.tracks:
        for note in track.notes:
            pitch_classes.append(note.pitch % 12)

    if not pitch_classes:
        return 0.0

    # Calculate histogram and probabilities
    hist = np.bincount(pitch_classes, minlength=12)
    probs = hist / np.sum(hist)

    # Calculate Shannon entropy (ignore zero probabilities)
    probs = probs[probs > 0]
    entropy = -np.sum(probs * np.log2(probs))

    return entropy
```

### B.1.2 EMPTY BEAT RATE (EBR)

**Definition:** The ratio of beat positions (quarter-note intervals) that contain no note onsets. This reflects the rhythmic density and "breathing space" of the arrangement.

$$EBR = \frac{\sum_{b=1}^{B} \mathbf{1}(\text{onset\_count}(b) = 0)}{B} \tag{9}$$

where $B$ is the total number of beats in the song, and $\mathbf{1}(\cdot)$ is the indicator function.

**Python Implementation:**

```python
def compute_empty_beat_rate(music_obj):
    # Gridify to 1 beat resolution (quarter note)
    grid = music_obj.to_pianoroll(resolution=1)

    # Check if any note exists in each beat
    has_notes = (grid.sum(axis=1) > 0)

    # Count empty beats
    empty_beats = len(has_notes) - has_notes.sum()

    return empty_beats / len(has_notes)
```

### B.1.3 GROOVE CONSISTENCY (GC)

**Definition:** Measures the stability of rhythmic patterns by calculating the similarity between adjacent measures. High consistency implies a stable rhythmic "groove."

$$GC = 1 - \frac{1}{M-1} \sum_{i=1}^{M-1} \frac{\text{Hamming}(\vec{v}_i, \vec{v}_{i+1})}{L} \tag{10}$$

where $M$ is the number of measures, $L$ is the number of time steps per measure, and $\vec{v}_i$ is the binary onset vector of measure $i$.

**Python Implementation:**

```python
def compute_groove_consistency(music_obj):
    # Get binary onset piano roll (resolution=4 per beat -> 16 per bar)
    pr = music_obj.to_pianoroll(resolution=4, binary=True)
    bar_length = 16
    num_bars = len(pr) // bar_length

    if num_bars < 2: return 0.0

    consistency_scores = []
    for i in range(num_bars - 1):
        # Extract binary onset vectors for adjacent bars
        v1 = (pr[i*bar_length : (i+1)*bar_length].sum(axis=1) > 0).astype
    (int)
        v2 = (pr[(i+1)*bar_length : (i+2)*bar_length].sum(axis=1) > 0).
    astype(int)

        # Calculate Hamming distance (element-wise mismatch)
        hamming_dist = np.mean(v1 != v2)
        consistency_scores.append(1 - hamming_dist)

    return np.mean(consistency_scores)
```

### B.1.4 SCALE CONSISTENCY (SC)

**Definition:** The maximum ratio of pitch classes that belong to a single standard major or minor scale. This validates the tonal clarity.

$$SC = \max_{k \in \mathcal{K}} \left( \frac{\sum_{n \in N} \mathbf{1}(\text{pitch}(n) \in S_k)}{|N|} \right) \tag{11}$$

where $N$ is the set of all notes, and $\mathcal{K}$ is the set of all 24 major/minor scales.

**Python Implementation:**

```
def compute_scale_consistency(music_obj):
    pitch_classes = [note.pitch % 12 for track in music_obj.tracks for
    note in track.notes]
    total_notes = len(pitch_classes)
    if total_notes == 0: return 0.0

    # Define scale intervals (Major and Minor)
    major_intervals = {0, 2, 4, 5, 7, 9, 11}
    minor_intervals = {0, 2, 3, 5, 7, 8, 10}

    max_in_scale = 0
    for root in range(12):
        for intervals in [major_intervals, minor_intervals]:
            scale_set = {(root + i) % 12 for i in intervals}
            count = sum(1 for pc in pitch_classes if pc in scale_set)
            max_in_scale = max(max_in_scale, count)

    return max_in_scale / total_notes
```

### B.1.5   POLYPHONY DEGREE (POLY.)

**Definition:** The average number of pitches being played simultaneously at any given time step (resolution: 1/16 note). It reflects the complexity of the texture.

$$PD = \frac{1}{T} \sum_{t=1}^{T} (\text{count of active pitches at time } t) \tag{12}$$

where $T$ is the total number of time steps.

**Python Implementation:**

```
def compute_polyphony(music_obj):
    # Get piano roll (resolution=4 per beat)
    pr = music_obj.to_pianoroll(resolution=4)
    # Count active notes at each time step
    active_notes = (pr > 0).sum(axis=1)
    # Return average
    return np.mean(active_notes)
```

### B.2   STRUCTURAL AND CONTROL METRICS

**Note:** In addition to standard metrics, we define the specific metrics used to evaluate controllability and structural form.

### B.2.1   CHORD HIT RATE (HIT RATE)

**Definition:** Measures the strict adherence of generated notes to the input chord constraints. A value of 1.0 implies all notes are chord tones.

$$H = \frac{\sum_{i=1}^{M} \sum_{n \in N_i} \mathbf{1}((n_{\text{pitch}} \bmod 12) \in \text{PC}(c_i))}{\text{Total Notes}} \tag{13}$$

where $N_i$ is the set of notes generated within the duration of chord $c_i$, and $\text{PC}(c_i)$ is the set of pitch classes forming chord $c_i$.

### B.2.2 SIMILARITY ERROR (SE)

**Definition:** Adapted from Museformer (Yu et al., 2022), this metric evaluates long-term structural integrity. It calculates the divergence between the structural similarity distributions of generated music ($\hat{L}$) and real human-made music ($L$).

$$SE = \frac{1}{T} \sum_{t=1}^{T} |\hat{L}_t - L_t| \tag{14}$$

where $L_t$ represents the average similarity between bar pairs with an interval of $t$.

## C SUBJECTIVE EVALUATION DETAILS

To ensure full transparency regarding our user study, we provide detailed profiles of the 15 professional evaluators . The professional group was recruited from top-tier music conservatories and university music departments. All participants possess at least 8 years of formal musical training.

Table 6: Demographic profiles of the 15 professional evaluators.

| ID | Group | Profession/Major | Academic Status | Training (Yrs) | Area of Expertise |
|----|-------|------------------|-----------------|----------------|-------------------|
| P01 | Theory | Composition | PhD Candidate | 18 | Traditional harmony, Counterpoint |
| P02 | Theory | Music Theory | Master Student | 15 | Schenkerian analysis, Form |
| P03 | Theory | Film Scoring | Freelance | 12 | Emotional guidance, Orchestration |
| P04 | Theory | Jazz Composition | Senior Undergrad | 10 | Complex extensions, Modulation |
| P05 | Theory | Solfège | Instructor | 20 | Aural training |
| P06 | Perf. | Piano Performance | Master Student | 16 | Classical literature, Touch |
| P07 | Perf. | Piano Performance | Senior Undergrad | 14 | Romantic repertoire |
| P08 | Perf. | Accompaniment | Professional | 12 | Keyboard harmony |
| P09 | Perf. | Pop/Jazz Keyboard | Senior Undergrad | 9 | Pop progressions, Rhythm |
| P10 | Perf. | Conducting | Senior Undergrad | 13 | Score reading, Balance |
| P11 | App. | Music Tech | Master Student | 8 | MIDI arrangement, Rendering |
| P12 | App. | Music Education | Master Student | 11 | Pedagogy, Aesthetics |
| P13 | App. | Musicology | PhD Candidate | 12 | Stylistic analysis |
| P14 | App. | Electronic Music | Indie Musician | 7 | Synthesizer textures |
| P15 | App. | Music Therapy | Senior Undergrad | 9 | Emotion perception |
| **Avg.** | – | – | – | **12.4** | – |

