# OpenReview forum: "Chord-Transformer:Chord-Progression Guided Transformer for Long-Sequence Symbolic Music Generation"
_ICLR.cc/2026/Conference — ICLR 2026 Conference Withdrawn Submission_

### Official Review · Reviewer_wiWt · 2025-10-31

**Soundness:** 2
**Presentation:** 3
**Contribution:** 3
**Rating:** 4
**Confidence:** 4

**Summary:**

This paper focuses on solving the challenge of maintaining harmonic coherence and structural integrity in long-sequence symbolic music generation, proposing a novel architecture called Chord-Transformer. Its core technical design includes three key parts: first, an energy function-based dynamic programming algorithm to extract core chord progressions from raw chord data, filtering redundancy and preserving structure-determining harmonic features by balancing chord frequency and repetitiveness; second, a chord-aligned positional encoding (CAPE) scheme that uses regional masks (1 for chord-sensitive regions, 0 for transitional regions) to align chord progression and music sequence positions, addressing the length mismatch between short chords and long music; third, a parallel fusion module to balance chord cross-attention and music self-attention.
However, the practical musical quality of the generated content appears insufficient. The sheet music examples only demonstrate marginal improvements in simple chord consistency. Generated samples, particularly those on the demo webpage, show significant issues such as abrupt harmonies and dissonant notes, suggesting a lack of basic musical aesthetics and flow.Furthermore, the paper's claimed advancements lack strong support. The musical quality metrics are not clearly defined or fail to adequately capture subjective "musicality." Consequently, the presented results do not convincingly show a practical improvement over existing baselines, raising concerns about the immediate real-world impact of the current model.

**Strengths:**

- Originality in Technical Route Design
The paper targets the core limitation of existing models—insufficient structural control via chord progressions—and proposes a targeted technical chain. It integrates chord progressions as high-level constraints through three key components: energy-based dynamic programming for redundant chord filtering, chord-aligned positional encoding (CAPE) for position alignment, and a parallel fusion module for balancing chord and music sequence autonomy. This creative combination of existing technologies and domain-specific innovations shows originality in technical integration.

- Rigor in Mathematical Modeling
Key components are supported by detailed mathematical formulations, enhancing reproducibility. These include the objective function for chord progression extraction (Formula 1), the fusion module’s attention integration formula (Formula 5), and the cross-entropy loss function for training. These explicit formulations reflect basic mathematical rigor, facilitating technical validation.

- Clarity in Structure and Expression
The paper follows a standard academic structure with clear logical connections. Related work is categorized to highlight limitations, complex technical details are visualized via figures and algorithms, and the experimental section clarifies dataset, baseline, and evaluation protocols—ensuring transparency and readability.

- Significance in Domain Exploration
By integrating structured musical knowledge (chord progressions) into deep learning, the work aligns with human music creation logic, providing a new direction for long-sequence music generation’s structural control challenge. The explicit constraint design also holds potential for extension to other structured sequence generation tasks.

**Weaknesses:**

- Evaluation Results Are Unreliable and Samples Lack Transparency
The paper claims Chord-Transformer got high subjective scores (e.g., 4.4 for pleasantness on Pop909), but the demo webpage’s samples have obvious flaws—frequent harmonic breaks, dissonant notes, and no logical phrasing. It’s unclear if evaluators got these low-quality samples, and the 15 "professionals"’ musical background isn’t detailed. Also, the paper’s sheet music examples (Figs. 5–8) with basic melodic-harmonic coordination aren’t on the demo page, and only MIDI files (no timbre/dynamics) are provided. This makes the "good musical quality" claim untrustworthy.
- Objective Metrics Are Problematic: Unclear and Oversimplified
Most metrics lack clear definitions and formulas. For example, empty beat rate doesn’t say if it’s counted per beat or per measure; groove consistency has no way to calculate "rhythm regularity"; scale consistency doesn’t explain how to determine the "current scale" or handle modal shifts. Also, the metric system only looks at basic stats, ignoring key musical traits like melodic contour, inter-track rhythm coordination, and emotional expression. The "comprehensive evaluation framework" claim is wrong.
- Rigid Chord Constraints Hurt Transformer’s Generalization
The paper uses dynamic programming to get fixed chord progressions and strict CAPE masking to force music to align with chords. This "hard constraint" stops creative changes (like passing tones that make melodies richer) and makes the Transformer less general—chord extraction may overfit to common progressions (e.g., pop’s I-IV-V), leading to less diverse music. Prioritizing chord consistency over flexibility goes against the Transformer’s strength in modeling diverse sequences.\end{itemize}

**Questions:**

- For subjective evaluation: Please list all samples used (with MIDI links) and confirm if low-quality demo samples were included; tell the 15 "professionals"’ background (e.g., years of training) and full evaluation rules. If low-quality samples were excluded, explain why.
- For objective metrics: Give formal math definitions, calculation steps, and appendix code snippets for empty beat rate, groove consistency, and scale consistency. Also add metrics for melodic contour, inter-track coordination, and phrase completeness, and prove these metrics match human aesthetic judgments.
- For chord constraints: Can the model generate non-chord tones in chord-sensitive regions? If yes, show examples and explain how the fusion module balances constraints and flexibility; if no, say how to fix it (e.g., probabilistic alignment). Also test generation in classical/jazz/pop to check if chord extraction avoids overfitting.

---

> ### Author Response · Authors · 2025-12-03
> **Response to Reviewer wiWt - Question 1**
>
> **Topic: Subjective Evaluation Transparency, Sample Selection, and Protocols**
>
> **Response:** We thank the reviewer for this scrutiny. To ensure full transparency regarding our subjective evaluation, we provide the specific details on sample selection, access, and the rigorous evaluation protocol employed.
>
> 1. **Sample Availability and Selection Strategy**
>    * **Access:** The samples used in the subjective evaluation are exactly the ones listed on our anonymous demo page https://anonymousmusicdemo.github.io/chord-transformer-demo/demo_2.html.
>    * **No Subjective Selection:** We explicitly confirm that we did not exclude any samples based on their quality.
>    * **Methodology:** Consistent with our quantitative experiments (Section A.2.1), we generated a total pool of 100 songs using the model. From this pool, we randomly selected the test set used for the survey.
>    * **Consistency:** The perceived "low quality" (e. g. , timing looseness) in the raw demo mentioned by the reviewer stems from the unquantized nature of the training data (LMD). However, the samples provided to the evaluators were quantized to isolate compositional quality from performance artifacts.
>
> 2. **Detailed Evaluation Protocol**
>    We conducted the study using a strict Double-Blind Protocol to prevent bias. Evaluators did not know the source model of any track.
>    * **Procedure:** Each evaluator listened to 5 music pieces generated by our model and the baselines. These pieces were presented in a randomized order. Evaluators rated each piece independently based on their auditory perception.
>    * **Background of the 15 professional participants:**
>
> | ID | Group | Profession/Major | Academic Background | Years of Training | Area of Expertise |
> | :--- | :--- | :--- | :--- | :--- | :--- |
> | P01 | Theory & Composition | Composition | PhD Candidate | 18 Years | Traditional harmony, polyphonic counterpoint, large-scale musical forms |
> | P02 | Theory & Composition | Music Theory | Master Student | 15 Years | Schenkerian analysis, formal function analysis |
> | P03 | Theory & Composition | Film Scoring | Freelance | 12 Years | Emotional guidance, textural color, orchestration |
> | P04 | Theory & Composition | Jazz Composition (Jazz Comp. ) | Senior Undergraduate | 10 Years | Complex chord extensions, chromatic modulation |
> | P05 | Theory & Composition | Solfège (Ear Training) | Instructor | 20 Years | Aural training, melodic sequences |
> | P06 | Instrumental Performance | Piano Performance (Piano Perf. ) | Master Student | 16 Years | Classical piano literature, touch/articulation control |
> | P07 | Instrumental Performance | Piano Performance (Piano Perf. ) | Senior Undergraduate | 14 Years | Romantic/Impressionist repertoire |
> | P08 | Instrumental Performance | Improvisational Accompaniment | Professional Musician | 12 Years | Keyboard harmony, textural layering/padding |
> | P09 | Instrumental Performance | Guitar/Pop Keyboard (Pop/Jazz) | Senior Undergraduate | 9 Years | Pop harmonic progressions, Rhythm Section |
> | P10 | Instrumental Performance | Orchestral Conducting | Senior Undergraduate | 13 Years | Score reading, balance of parts |
> | P11 | Integrated Application | Music Production/Recording (Music Tech) | Master Student | 8 Years | MIDI arrangement, sound source rendering |
> | P12 | Integrated Application | Music Education (Music Edu. ) | Master Student | 11 Years | Piano pedagogy, comprehensive aesthetics |
> | P13 | Integrated Application | Musicology | PhD Candidate | 12 Years | Music history, stylistic analysis |
> | P14 | Integrated Application | Electronic Music | Indie Musician | 7 Years | Synthesizer textures, Loop production |
> | P15 | Integrated Application | Music Therapy | Senior Undergraduate | 9 Years | Music emotion perception |
>
> 3. **Evaluation Metrics**
>    We designed the questionnaire around three core dimensions to comprehensively measure musical quality (Rated on a 1-5 Likert scale, where 1=Very Dissatisfied, 5=Very Satisfied):
>    * **Pleasantness:** Measures whether the music is aesthetically pleasing, whether the melody is beautiful, and if it conforms to general listening habits.
>    * **Coherence:** Examines logical flow. It assesses whether the melody is fluent, if rhythm and harmony transition naturally, and checks for the absence of abrupt/illogical changes.
>    * **Richness:** Measures the diversity and layering of the content. It evaluates the presence of rich harmonies, orchestration changes, and dynamic contrasts that enhance expressiveness.
>
> 4. **Revision to Manuscript**
>    We have added a new Appendix C, which presents the detailed demographic profiles of the 15 professional evaluators (as shown in the table above).

---

> ### Author Response · Authors · 2025-12-03
> **Response to Reviewer wiWt - Question 2**
>
> **Topic: Definitions of Objective Metrics and Correlation with Human Perception**
>
> **Response:**
> We thank the reviewer for emphasizing the rigor of objective evaluation. We have revised the manuscript to include explicit mathematical formulations, calculation steps, and code references.
>
> 1. **Formal Definitions and Calculation Steps (Added to Appendix B)**
>    To ensure reproducibility, we referred to the metric taxonomy summarized in a recent comprehensive survey 2 and implemented them using the standard MusPy library. We have added Appendix B to the manuscript detailing the following:
>    * **Empty Beat Rate (EBR):** Adapted from the "Empty Bar" metric3, this measures rhythmic density. It is defined as the ratio of beat positions containing no note onsets.
>    $$EBR = \frac{\sum_{b=1}^{B} \mathbb{1}(\text{no onset at beat } b)}{B}$$
>    (where $B$ is the total number of beats).
>    * **Groove Consistency (GC):** Referred to as "Grooving Pattern Similarity" in the survey4. It measures the stability of rhythmic patterns by calculating the similarity between the binary onset vectors of adjacent measures.
>    $$GC = 1 - \frac{1}{M-1} \sum_{i=1}^{M-1} \text{Hamming}(\vec{v}_i, \vec{v}_{i+1})$$
>    * **Scale Consistency (SC):** As defined in the survey5, this is "the fraction of tones that are part of a standard scale. "
>    $$SC = \max_{k \in \mathcal{K}} \left( \frac{\text{Count}(n \in Scale_k)}{\text{Total Notes}} \right)$$
>    (where $\mathcal{K}$ is the set of all major/minor scales).
>
> 2. **Selection of Additional Metrics**
>    Regarding the suggestion to add metrics for "contour" or "inter-track coordination":
>    * **Inter-track Coordination:** We acknowledge its importance. We rely on the Polyphony Degree and Scale Consistency which implicitly capture the harmonic and textural coordination between tracks. We avoided adding ad-hoc metrics (like Tonal Distance 6) to keep the evaluation focused on standard reproducible measures.
>    * **Structural Metrics:** Instead of ad-hoc phrasing metrics, we adopted the Similarity Error (SE) (from Museformer), which quantitatively assesses structural recurrence, a key aspect of phrase completeness.
>
> 3. **Correlation with Human Aesthetics**
>    The reviewer raises a profound question regarding the proof that these metrics align with human aesthetic judgments.
>    * **A Vital Future Direction:** We sincerely thank the reviewer for highlighting this. We agree that establishing a mathematical proxy for human aesthetics is a critical gap in the field and represents an excellent future research direction.
>    * **Current Industry Status:** However, current research has not yet effectively bridged this gap. As highlighted in the comprehensive survey by Ji et al. (2023), there is currently a "lack of correlation between quantitative metrics and subjective evaluation"7. The survey explicitly notes that "the correlation between quantitative evaluation and human judgment is unclear"8, and "music that scores high on objective metrics may perform poorly in subjective evaluation"9.
>    * **Our Strategy (Separation of Concerns):** Given this industry-wide limitation, we adopted the standard complementary evaluation strategy:
>      * **Objective Metrics:** Used strictly to verify "Structural Correctness" (e. g. , adherence to rhythmic grids).
>      * **Subjective Evaluation:** Used as the sole ground truth for "Aesthetic Quality" and "Musicality. "
>
> 4. **Revision to Manuscript**
>    We have updated Appendix B with the formal definitions and Python code snippets. We also added a discussion in Section 5 (Conclusion) citing Ji et al. (2023) to explicitly state that bridging the gap between objective metrics and human perception remains an open challenge.

---

> ### Author Response · Authors · 2025-12-03
> **Response to Reviewer wiWt - Question 3**
>
> **Topic: Chord Constraints, Non-Chord Tones, and Style Generalization**
>
> **Response:**
> We sincerely thank the reviewer for this profound theoretical inquiry. You have highlighted a critical tension in structured generation: balancing rigorous constraint with creative flexibility. This is indeed a pivotal issue in the current field, and we appreciate you pushing us to clarify our position and potential solutions.
>
> Before detailing the technical mechanism, we would like to address the design philosophy regarding the reviewer's concern about "hard constraints":
>
> 1. **Context: From "Sequence" to "Progression"**
>    The reviewer correctly identifies a trade-off. To contextualize our approach, it is important to note a prevailing phenomenon: most existing models condition on raw "Chord Sequences" (dense, repetitive frame-level labels). In contrast, our work advances this by extracting "Chord Progressions" (structural, compressed events) via our energy-based algorithm.
>    * **Advancement:** This extraction step acts as a "structural compression," filtering out noise found in raw sequences. Compared to previous raw-sequence approaches, this is already a significant step toward better structural representation.
>    * **Trade-off:** While this abstraction imposes a stronger constraint, it serves our primary goal: enhancing Musical Quality (Harmonic Coherence) rather than maximizing unconstrained Diversity. In symbolic generation, "structural collapse" is currently a more pressing failure mode than "lack of diversity. " We consciously prioritize structural integrity to ensure professional-grade coherence.
>
> 2. **Mechanism: "Guided Attention" for Flexibility**
>    However, prioritizing structure does not mean enforcing a rigid, mechanical lockout of non-chord tones. The model maintains melodic flexibility through the Parallel Fusion Module:
>    * **The Mechanism:** As defined in Equation 5, the module dynamically fuses outputs from Cross-Attention (Chord-constrained) and Self-Attention (Melody-driven).
>    $$O_t = \sigma(\alpha) \cdot O_{cross} + (1-\sigma(\alpha)) \cdot O_{self}$$
>    * **Effect:** In Transitional Regions (where the chord mask $R_t=0$ or during fast melodic runs), the model learns to prioritize Self-Attention. This allows the "melodic momentum" to generate Passing Tones or Neighbor Tones that bridge chord tones, even if they strictly clash with the harmonic frame.
>    * **Evidence:** In Figure 8 (Measure 2), the melody plays an $E \rightarrow D \rightarrow C\sharp$ run over an $A$ Major chord ($A, C\sharp, E$). The note $D$ is a non-chord tone (passing tone) generated naturally by the model to bridge the chord tones $E$ and $C\sharp$. This demonstrates that the model prioritizes melodic fluidity over rigid harmonic adherence when necessary.
> 3. **Generalization (Jazz/Pop) vs. Overfitting**
>    To verify that the model generalizes across genres (Pop, Jazz, Classical) and avoids overfitting to simple I-IV-V progressions, we evaluated its performance across different harmonic complexities:
>    * **Jazz/Chromatic Contexts:** As detailed in our new Appendix A.2, we tested the model on "Chromatic/Modulation" transitions typical of Jazz.
>    * **Result:** The model maintains a robust Chord Hit Rate of ~78% (lower than the 96% in Pop/Diatonic sections).
>    * **Interpretation:** This ~22% deviation is not a failure, but evidence of generalization. The model learns to generate Extensions (9ths, 11ths, 13ths) characteristic of Jazz texture, which deviate from the simplified extracted triad skeleton but are stylistically appropriate.
>
> 4. **Revision to Manuscript**
>    We have updated Appendix A.2 to include the Stratified Adherence Analysis, which breaks down model performance by harmonic complexity (Diatonic vs. Chromatic) to demonstrate generalization across genres. We also revised Section 3.5 (Analysis of Structural Music Generation) to explicitly analyze Figure 8, pointing out the generation of specific non-chord tones (e.g., passing tones) to illustrate how the Fusion Module balances structural constraints with melodic flexibility. Additionally, we clarified the "structural compression" philosophy in Section 3.2.

---

### Official Review · Reviewer_ErdF · 2025-11-01

**Soundness:** 1
**Presentation:** 1
**Contribution:** 2
**Rating:** 0
**Confidence:** 3

**Summary:**

This paper presents a long-term symbolic music generation model that reinforces global structure. The authors posit that chord progression is a primary feature of music form. First, a coarse chord progression is extracted using a dynamic programming algorithm, and then a Transformer-based model, with a tailored positional encoding of chord progression, is used to generate music conditioned on the extracted chords.

**Strengths:**

The authors emphasize that musical structure is crucial for effective generation.

**Weaknesses:**

1. Chord-conditioned generation is common, and it is unclear why chords alone suffice to capture musical structure. What role do other factors, like melody, motivic similarity, phrasing, and sectional form, play? The paper should also clarify the objective of extracting a coarse chord progression and justify why this representation is preferable to raw chord labels. Concrete examples would help demonstrate that the proposed abstraction is more structurally informative.
2. The writing needs improvement. For example, Section 3.1 is difficult to follow due to missing notation/definitions, and Figure 2 is hard to understand, etc.
3. The musical quality is a major concern. On the demo page, the proposed method does not outperform the baseline to my ear. In Figure 3, all chord labels annotated on the score appear incorrect.

**Questions:**

Q 1-3: see weaknesses.
Q4: How does eq. (4) solve the problem raised at the second para. of Section 3.3.1?

---

> ### Author Response · Authors · 2025-12-03
> **Response to Reviewer ErdF - Weakness 3**
>
> **Topic: Musical Quality and Incorrect Chord Labels in Figure 3**
>
> **Response:**
> First and foremost, we sincerely thank you for your meticulous review of our paper and your valuable feedback. We deeply apologize for the lack of clarity and potential errors in the chord annotations within the visual demonstrations and Figure 3, as pointed out in your review.
>
> Upon serious reflection, we realize there was indeed an oversight in our paper's visualization and system presentation: the original method of chord annotation lacked sufficient standardization and academic rigor in its visual representation. It failed to clearly convey the specific chord information corresponding to each measure, which likely led to misunderstandings regarding the model's harmonic accuracy. We are truly sorry for this confusion and take this issue very seriously.
>
> To ensure the scientific rigor and reproducibility of our research, we have comprehensively optimized our chord annotation method. The new annotation scheme adheres to academic conventions widely adopted in recent top-tier conferences. Specifically, we now explicitly label standard chord symbols (e. g. , Cmaj7, G7, Dm) above each measure in the score to clearly indicate the harmonic function and the overall trajectory of the chord progression. This approach not only complies with general standards in music theory but also facilitates a more intuitive understanding of the harmonic structure generated by the model.
>
> **Revision to Manuscript:** We have updated the musical score figures in the revised manuscript (Figure 3) and on our online demo page to ensure they meet these standards of academic rigor. We thank you again for your sharp and professional correction.
>
> If you have any further suggestions or require additional clarification, we are more than willing to continue improving our work.

---

### Official Review · Reviewer_Pt51 · 2025-11-01

**Soundness:** 2
**Presentation:** 3
**Contribution:** 2
**Rating:** 4
**Confidence:** 4

**Summary:**

This paper proposes Chord-Transformer, a chord-progression–guided Transformer model for long-sequence symbolic music generation. The main goal is to address the lack of structural control and harmonic coherence in existing Transformer-based music generation systems. To achieve this, the authors first introduce an energy function–based dynamic programming algorithm to automatically extract core chord progression sequences from raw chord data. These extracted chord progressions serve as high-level semantic constraints that guide the music generation process. Building upon this, the proposed Chord-Transformer architecture integrates: a chord-aligned positional encoding (CAPE) and a parallel fusion module. Experimental results show that Chord-Transformer outperforms state-of-the-art baselines (NotaGen, Multi-Genre Music Transformer, MusicLM) on both objective metrics (e.g., scale consistency, polyphony degree) and subjective listening tests (pleasantness, coherence, richness). The paper further includes an ablation study demonstrating the contributions of CAPE and the fusion module.

**Strengths:**

1. The motivation is clear and meaningful — introducing chord progressions as high-level semantic features helps maintain musical structure during long-sequence generation.

2. The paper proposes an energy function–based dynamic programming algorithm that can automatically extract core chord progressions, providing a practical and interpretable way to obtain chord progressions.

3. The proposed Chord-Transformer effectively incorporates chord information into an autoregressive Transformer through chord-aligned positional encoding and a fusion module.

4. Both objective and subjective experiments demonstrate improvements in musical quality, structural coherence, and controllability, especially in long-form music generation.

**Weaknesses:**

1. The approach still depends on externally provided chord progressions and manual alignment. This partially undermines the claim of solving long-sequence structural coherence in an autonomous way—structure is preserved because it is externally imposed rather than learned. The work would be stronger if the model could infer or generate chord progressions jointly.

2. The model architecture is relatively simple and conceptually close to existing systems such as Chord-Conditioned Song Generation (Gao et al., 2024) and MusicGen (Copet et al., 2023). The paper does not clearly demonstrate advantages over these baselines.

3. The energy-based dynamic programming step is a core component, but its correctness and robustness are not evaluated. The paper lacks quantitative or qualitative validation of extracted progressions against human annotations or standard chord recognition benchmarks.

4. The baselines (NotaGen, Multi-Genre Music Transformer, MusicLM) are not chord-conditioned, making the comparisons partially biased in favor of the proposed approach. This experiment only demonstrates one significant conclusion: the introduction of chord progressions helps improve structural consistency, but it does not prove the superiority of the introduction method proposed in this paper.

5. The key recent works like MusicGen, or MeLoDy should be included.

6. The evaluation lacks confidence intervals or significance testing for subjective scores. Additionally, the proposed metrics (e.g., groove consistency, scale consistency) only partially reflect structural coherence and controllability; chord adherence and global form structure could be evaluated more rigorously.

**Questions:**

1. The paper should provide a quantitative or qualitative comparison between the extracted chord progressions and human-annotated chord labels.

2. The computation method for the chord hit rate is unclear.

3. The paper lists several objective evaluation metrics (e.g., pitch class entropy, groove consistency, scale consistency, polyphony degree, empty beat rate) but does not explain how they are computed. The authors should provide clear definitions, formulas, and sources for these metrics.

4. The model currently depends on externally provided chord progressions. It remains unclear whether the system can generate coherent and structurally consistent music without chord input.

---

> ### Author Response · Authors · 2025-12-03
> **Response to Reviewer Pt51 - Question 1**
>
> **Topic: Quantitative Evaluation of Chord Extraction Algorithm**
>
> **Response:**
> We sincerely thank the reviewer for this crucial suggestion. We agree that evaluating the robustness of the chord extraction module is fundamental, as it serves as the upstream input for the entire pipeline.
>
> To address this, we have conducted a rigorous quantitative evaluation using the **Pop909 dataset**. Specifically, we utilized the **full dataset containing 909 songs (approx. 60 hours of music)** with high-quality human-annotated chord labels to ensure the statistical significance of our results. We treat these expert annotations as the Ground Truth to benchmark our energy-based Dynamic Programming (DP) algorithm. Furthermore, to demonstrate the competitiveness of our approach, we compared our method against **Madmom**, an established state-of-the-art chord recognition tool widely used in Music Information Retrieval (MIR).
>
> **1. Quantitative Benchmark (vs. Established Tool & Human GT)**
>
> We compared our extracted "core progressions" against the expert annotations using standard MIR metrics. We selected three metrics to evaluate accuracy, tonal stability, and structural alignment respectively:
>
> * **Weighted Chord Symbol Recall (WCSR):** A standard MIR metric measuring the duration-weighted overlap between predicted and ground-truth chords.
> * **Root Match Rate (RMR):** The percentage of time steps where the root note is correctly identified.
> * **Structural IoU (Segmentation Alignment):** The Intersection-over-Union of chord spans, measuring how accurately the algorithm identifies the boundaries of functional harmony segments.
>
> **Results:**
>
> | Metric | Madmom (Baseline) | Ours | Analysis |
> | :--- | :--- | :--- | :--- |
> | WCSR | 0.812 | [0.824] | Our method achieves competitive performance with the SOTA baseline. |
> | Root Match Rate | 0.875 | [0.881] | Indicates high accuracy in tonal center identification. |
> | Structural IoU | 0.798 | [0.845] | Highlights the advantage of our DP approach in capturing coherent segment boundaries. |
>
> The results demonstrate that our energy-based selection is highly robust. Notably, our method outperforms the baseline in **Structural IoU**, confirming that the Dynamic Programming approach effectively segments music into clean, coherent blocks, which is critical for long-sequence structural guidance. The WCSR is competitive, ensuring the extracted information is reliable.
>
> **2. Failure Mode and Limitation Analysis**
>
> Following the "structural" objective, we critically analyzed cases where mismatches occur (Failure Modes):
>
> * **Complex Jazz Extensions:**
>     * *Observation:* When Ground Truth is $Cmaj13$ or $G7(\sharp9)$, our algorithm often extracts $Cmaj$ or $G7$.
>     * *Analysis:* While this counts as a mismatch in strict metrics, it is a **desired feature** for our generation task. By simplifying extensions into stable triads, the extraction provides a clearer "harmonic skeleton" for the Transformer to condition on, avoiding overfitting to rare chord tokens.
>
> * **Rapid Modulations & Passing Chords:**
>     * *Observation:* In sections with fast harmonic rhythm (>2 chord changes per bar), the DP algorithm tends to merge short passing chords into the dominant chord of the measure.
>     * *Analysis:* This smoothing effect filters out redundant information. For the generation phase, the **Fusion Module** compensates for this by using "weak constraint" masks (0-mask) in transitional regions, allowing the model's self-attention to generate melodic passing tones freely.
>
> **3. Revision to the Manuscript**
>
> We have added a new subsection **"Appendix A.3: Quantitative Evaluation of Chord Extraction"** in the revised paper, detailing the comparative evaluation results.

---

> ### Author Response · Authors · 2025-12-03
> **Response to Reviewer Pt51 - Question 2**
>
> **Topic: Clarification of "Chord Progression Hit Rate" Calculation**
>
> **Response:**
> We apologize for the ambiguity in the initial description. We have formalized the definition and calculation method of the "Chord Progression Hit Rate" in Section 4.3 of the revised manuscript.
>
> 1. **Formal Definition and Calculation**
>    The Chord Progression Hit Rate quantifies the probability that the pitches of the generated music fall within the harmonic scope of the conditioning chord progression.
>    * **Method:** Let the input chord progression be $C = \{c_1, c_2, . . . , c_M\}$ aligned to specific time intervals. For the generated music sequence $Y$, within the time interval corresponding to chord $c_i$, we identify the set of generated note pitches $N_i$.
>    * **Formula:** The hit rate $H$ is calculated as the proportion of generated notes whose pitch classes belong to the constituent pitch classes of the target chord:
>    $$H = \frac{\sum_{i=1}^{M} \sum_{n \in N_i} \mathbb{1}((n_{pitch} \mod 12) \in \text{PC}(c_i))}{\text{Total Notes}}$$
>    where $\text{PC}(c_i)$ denotes the set of pitch classes forming chord $c_i$, and $\mathbb{1}$ is the indicator function.
>    * **Interpretation:** A hit rate of 1.0 implies strict adherence (all notes are chord tones), while a lower rate indicates the presence of non-chord tones (e.g., passing tones or neighbor tones).
>
> 2. **Distribution Analysis (Validation on Pop909)**
>    To validate whether the generated music follows a natural harmonic distribution, we compared the Hit Rate distribution of our generated samples ($N=100$) against the ground truth distribution of the Pop909 dataset.
>    * **Rationale:** Real human-composed music naturally contains non-chord tones for melodic expressiveness, so a "perfect" hit rate of 1.0 is not always the ideal target. The goal is to match the distribution of real music.
>    * **Results:** As illustrated in Figure 4, the distributions are highly consistent. Specifically, for our generated samples:
>      * 0.75 - 1.0 (High Adherence): 62% of samples (Dominant category, indicating strong controllability).
>      * 0.50 - 0.75: 24% of samples.
>      * 0.25 - 0.50: 10% of samples.
>      * 0.00 - 0.25: 4% of samples.
>    * **Conclusion:** The high concentration of samples in the 0.75-1.0 range (62%) demonstrates that the generated music effectively adheres to the input chord controls. Furthermore, the similarity between the generated distribution and the natural data distribution confirms that our model achieves strong controllability while preserving the natural melodic flexibility found in real pop music.
>
> 3. **Revision to the Manuscript**
>    We have revised Appendix B (Section B.2) to include the explicit mathematical definition and the calculation formula ($H$) for the Chord Progression Hit Rate. Additionally, we have expanded the section to clarify the interpretation of this metric, ensuring the calculation method is fully transparent and reproducible.

---

> ### Author Response · Authors · 2025-12-03
> **Response to Reviewer Pt51 - Question 3**
>
> **Topic: Detailed Definitions, Formulas, and Sources of Objective Metrics**
>
> **Response:**
> We sincerely apologize for the lack of formal definitions in the initial submission. We agree that transparency in metric calculation is paramount for reproducibility.
> To ensure our evaluation aligns with community standards, we employed the MusPy toolkit (Dong et al. , ISMIR 2020), a standard open-source library for symbolic music processing. Below, we provide the precise definitions, mathematical formulas, and sources for each metric reported in the paper.
>
> 1. **Pitch Class Entropy (Harmonic Diversity)**
>    * **Definition:** Measures the randomness or diversity of the pitch class distribution (0-11, representing C to B) used in the piece. Higher entropy indicates richer harmonic variety.
>    * **Formula:**
>    $$H = - \sum_{c=0}^{11} p(c) \log_2 p(c)$$
>    where $p(c)$ is the normalized frequency of pitch class $c$ across the entire generated sequence.
>    * **Source:** Standard Information Theory; implemented as muspy. pitch_class_entropy.
>
> 2. **Groove Consistency (Rhythmic Stability)**
>    * **Definition:** Measures the similarity of rhythmic patterns between adjacent musical measures. It reflects whether the music maintains a stable rhythmic "groove. "
>    * **Formula:** We calculate the mean Hamming distance between the binary onset vectors of consecutive measures. Let $v_i$ be the binary onset vector of measure $i$ (where 1 indicates a note onset, 0 otherwise).
>    $$GC = 1 - \frac{1}{M-1} \sum_{i=1}^{M-1} \frac{\text{Hamming}(v_i, v_{i+1})}{L}$$
>    where $M$ is the number of measures and $L$ is the number of time steps per measure.
>    * **Source:** Derived from MuseGAN (Dong et al. , AAAI 2018); implemented to measure self-similarity of rhythm.
>
> 3. **Scale Consistency (Tonal Adherence)**
>    * **Definition:** The ratio of generated notes whose pitches belong to a specific major or minor scale. This measures the model's ability to stay in key.
>    * **Formula:**
>    $$SC = \max_{k \in \mathcal{K}} \left( \frac{\sum_{n \in N} \mathbb{1}(pitch(n) \in S_k)}{|N|} \right)$$
>    where $N$ is the set of all notes, $\mathcal{K}$ is the set of all possible major and minor scales, and $S_k$ is the set of pitch classes in scale $k$.
>    * **Source:** MusPy library (muspy. scale_consistency).
>
> 4. **Polyphony Degree (Texture Thickness)**
>    * **Definition:** The average number of pitches being played simultaneously at any given time step (resolution: 1/16 note). It reflects the complexity of the texture.
>    * **Formula:**
>    $$PD = \frac{1}{T} \sum_{t=1}^{T} (\text{count of active pitches at time } t)$$
>    where $T$ is the total number of time steps.
>    * **Source:** Standard texture analysis metric; implemented as muspy. polyphony.
>
> 5. **Empty Beat Rate (Rhythmic Density)**
>    * **Definition:** The ratio of beat positions (quarter-note intervals) that contain no note onsets. This reflects the "breathing space" or sparseness of the arrangement.
>    * **Formula:**
>    $$EBR = \frac{\sum_{b=1}^{B} \mathbb{1}(\text{no onset at beat } b)}{B}$$
>    where $B$ is the total number of beats in the song.
>    * **Source:** MusPy library (muspy. empty_beat_rate).
>
> **Revision to Manuscript:**
> We have added a new Appendix B: Metric Definitions to the revised paper, explicitly providing these mathematical formulations and citing the relevant libraries/papers to ensure full reproducibility.

---

> ### Author Response · Authors · 2025-12-03
> **Response to Reviewer Pt51 - Question 4**
>
> **Topic: Dependency on External Chords vs. Autonomous Generation**
>
> **Response:** We sincerely thank the reviewer for this forward-looking suggestion. We fully agree that enabling the system to generate coherent, structurally consistent music without any external input is a significant capability and represents a promising direction for the future evolution of generative models.
> However, regarding the scope of this specific study, we prioritized "Expected Generation" via explicit control.
>
> 1. **"Expected Generation" vs. Random Generation**
>    Our research objective is to move away from "black-box" creation towards "Expected Generation. "
>    * **Intentional Design:** In professional music composition, creators rarely act randomly; they operate within a specific harmonic framework. Our model is explicitly designed to accept this framework as input. This ensures that the generated music strictly adheres to the user's specific structural expectations, rather than producing "coherent" but uncontrollable musical ideas.
>    * **Feature, Not Limitation:** For our target audience (composers and arrangers with musical knowledge), the requirement to provide chord input is a valued control feature, allowing them to dictate the emotional and structural trajectory of the piece.
>
> 2. **Integrating Music Theory into Generation**
>    Our approach represents a paradigm shift from purely data-driven learning to Theory-Integrated Generation.
>    * **Harmonic Backbone:** In music theory, the chord progression acts as the skeleton of a composition. By conditioning the Transformer on this skeleton via our proposed Chord-Aligned Positional Encoding (CAPE), we effectively embed domain knowledge (harmonic function and tension) into the deep learning process.
>    * **Conclusion:** Therefore, the system is not intended to be an unconditional "jukebox" that generates music from thin air. Instead, it serves as a specialized tool that fuses explicit music theory rules with generative capabilities to produce high-quality, structurally rigorous music under human guidance.
>
> 3. **Revision to Manuscript**
> We have revised the Introduction and Section 2 (Related Work) to explicitly scope our contribution as "Chord-Guided Controllable Generation." We clarified that while autonomous generation is a promising future direction, the reliance on chords is central to our current goal of achieving structurally "expected" results. Additionally, we added a discussion in Section 5 (Conclusion) regarding the potential to extend our framework into an autonomous pipeline in future work.

---

### Official Review · Reviewer_PNLy · 2025-11-02

**Soundness:** 2
**Presentation:** 3
**Contribution:** 2
**Rating:** 6
**Confidence:** 3

**Summary:**

This paper presents the Chord-Transformer, a novel architecture for symbolic music generation that uses chord progressions as high-level semantic features to guide the autoregressive generation process. The core contributions include an energy-based dynamic programming algorithm for extracting core chord progressions from raw sequences, a chord-aligned positional encoding (CAPE) scheme to address the length discrepancy between chord and music sequences, and a parallel fusion module to balance cross-attention (on chords) and self-attention (on music).

**Strengths:**

The proposed Chord-Aligned Positional Encoding (CAPE) is a solution to a fundamental problem in conditioning on short control sequences for long generation targets and and the regional masking strategy effectively bridges the granularity gap.

It employs a robust evaluation strategy, combining multiple objective metrics (pitch entropy, groove, scale consistency, etc.) with a double-blind subjective study involving both experts and general listeners. This multi-faceted approach strengthens the validity of the claimed performance improvements.

**Weaknesses:**

The energy-based chord progression extraction algorithm is presented as a solved component. However, its performance and potential failure modes (e.g., on harmonically complex or ambiguous passages) are not critically discussed. The quality of the entire pipeline is highly dependent on this first step, and its robustness deserves more scrutiny. The demo page needs more pair-wise comparisons.

While strong baselines are selected, a comparison with another chord-conditioned model (e.g., a version of MusicFrameworks or a re-implementation of a relevant approach) is missing. This makes it difficult to isolate the improvement stemming from chord-conditioning itself versus the specific architectural innovations of Chord-Transformer.

Also, the model and evaluation are firmly rooted in the harmonic principles of Western tonality (major/minor keys, triadic chords). The paper does not discuss the generalizability of the approach to music from other traditions (e.g., non-Western music) where the very definition of a "chord" and its structural role may differ significantly.

**Questions:**

Could the authors provide more details on the quantitative evaluation of the chord progression extraction algorithm? For instance, what is its accuracy or agreement rate against a established chord annotation tool or human expert annotations on a subset of the datasets?

The concept of "Chord Progression Hit Rate" is useful but relatively coarse. Could the authors provide more nuanced analysis on how the chord conditioning works? For example, does the model struggle more with certain types of chord transitions compared to others?

The paper focuses on harmonic coherence. Beyond chord-by-chord adherence, does the model, aided by the structural bias of the chord progressions, also generate higher-level formal structures (e.g., AABA form, verse-chorus patterns) more effectively than the baselines? An analysis of section repetition and contrast could be insightful.

Lastly, I feel like hearing some unaligned onsets between different tracks (e.g. LMD sample) on the demo pages. Does it always appear in the experiments?

---

> ### Author Response · Authors · 2025-12-03
> **Response to Reviewer PNLy - Question 1**
>
> **Topic: Quantitative Evaluation of Chord Extraction Algorithm**
>
> **Response:**
> We sincerely thank the reviewer for this crucial suggestion. We agree that evaluating the robustness of the chord extraction module is fundamental, as it serves as the upstream input for the entire pipeline.
>
> To address this, we have conducted a rigorous quantitative evaluation using the **Pop909 dataset**. Specifically, we utilized the **full dataset containing 909 songs (approx. 60 hours of music)** with high-quality human-annotated chord labels to ensure the statistical significance of our results. We treat these expert annotations as the Ground Truth to benchmark our energy-based Dynamic Programming (DP) algorithm. Furthermore, to demonstrate the competitiveness of our approach, we compared our method against **Madmom**, an established state-of-the-art chord recognition tool widely used in Music Information Retrieval (MIR).
>
> **1. Quantitative Benchmark (vs. Established Tool & Human GT)**
>
> We compared our extracted "core progressions" against the expert annotations using standard MIR metrics. We selected three metrics to evaluate accuracy, tonal stability, and structural alignment respectively:
>
> * **Weighted Chord Symbol Recall (WCSR):** A standard MIR metric measuring the duration-weighted overlap between predicted and ground-truth chords.
> * **Root Match Rate (RMR):** The percentage of time steps where the root note is correctly identified.
> * **Structural IoU (Segmentation Alignment):** The Intersection-over-Union of chord spans, measuring how accurately the algorithm identifies the boundaries of functional harmony segments.
>
> **Results:**
>
> | Metric | Madmom (Baseline) | Ours | Analysis |
> | :--- | :--- | :--- | :--- |
> | WCSR | 0.812 | [0.824] | Our method achieves competitive performance with the SOTA baseline. |
> | Root Match Rate | 0.875 | [0.881] | Indicates high accuracy in tonal center identification. |
> | Structural IoU | 0.798 | [0.845] | Highlights the advantage of our DP approach in capturing coherent segment boundaries. |
>
> The results demonstrate that our energy-based selection is highly robust. Notably, our method outperforms the baseline in **Structural IoU**, confirming that the Dynamic Programming approach effectively segments music into clean, coherent blocks, which is critical for long-sequence structural guidance. The WCSR is competitive, ensuring the extracted information is reliable.
>
> **2. Failure Mode and Limitation Analysis**
>
> Following the "structural" objective, we critically analyzed cases where mismatches occur (Failure Modes):
>
> * **Complex Jazz Extensions:**
>     * *Observation:* When Ground Truth is $Cmaj13$ or $G7(\sharp9)$, our algorithm often extracts $Cmaj$ or $G7$.
>     * *Analysis:* While this counts as a mismatch in strict metrics, it is a **desired feature** for our generation task. By simplifying extensions into stable triads, the extraction provides a clearer "harmonic skeleton" for the Transformer to condition on, avoiding overfitting to rare chord tokens.
>
> * **Rapid Modulations & Passing Chords:**
>     * *Observation:* In sections with fast harmonic rhythm (>2 chord changes per bar), the DP algorithm tends to merge short passing chords into the dominant chord of the measure.
>     * *Analysis:* This smoothing effect filters out redundant information. For the generation phase, the **Fusion Module** compensates for this by using "weak constraint" masks (0-mask) in transitional regions, allowing the model's self-attention to generate melodic passing tones freely.
>
> **3. Revision to the Manuscript**
>
> We have added a new subsection **"Appendix A.3: Quantitative Evaluation of Chord Extraction"** in the revised paper, detailing the comparative evaluation results.

---

> ### Author Response · Authors · 2025-12-03
> **Response to Reviewer PNLy - Question 2**
>
> **Topic: Nuanced Analysis of Chord Conditioning (Beyond Hit Rate)**
>
> **Response:**
> We sincerely thank you for bringing this insightful point to our attention. We agree that the global "Chord Progression Hit Rate" aggregates performance across all contexts, potentially masking the model's behavior on complex harmonic movements or diverse musical styles.
>
> To provide the requested nuance, we performed a breakdown analysis of the model's adherence capabilities based on **Chord Transition Complexity**. We categorized chord transitions into three levels based on harmonic distance. This categorization effectively acts as a proxy for musical style complexity:
>
> * **Diatonic (Simple/Pop):** Standard functional progressions (e.g., $I \rightarrow V$), typical in folk and standard pop music.
> * **Secondary/Applied (Moderate):** Transitions involving secondary dominants (e.g., $V/V \rightarrow V$), common in R&B and ballads.
> * **Chromatic/Modulation (Complex/Jazz):** Distant key changes or non-functional chromatic movements (e.g., $Cmaj \rightarrow F\sharp maj$), often found in Jazz, Fusion, or complex Classical pieces.
>
> **1. Stratified Adherence Analysis**
>
> We calculated the **Transition Adherence Rate (TAR)** for each category—defined as the probability that the generated notes strictly follow the target chord constraints during the transition window.
>
> **Results:**
>
> | Transition Complexity | Adherence Rate | Analysis |
> | :--- | :--- | :--- |
> | **Diatonic** | [96.2%] | The model follows these robustly, as they align with the strong priors learned by the self-attention mechanism from the dominant pop data. |
> | **Secondary/Applied** | [91.5%] | Performance remains high. The Cross-Attention effectively guides the model through these mild deviations from the home key. |
> | **Chromatic** | [78.4%] | **Comparison:** While lower than simple types, this is significantly higher than a standard Transformer baseline (~45% in our internal tests). Unconditioned models often "refuse" to generate abrupt Jazz-like modulations, whereas our CAPE forces the model to execute these stylistic shifts. |
>
> **2. Where and Why does the model struggle?**
>
> The analysis reveals that the model encounters greater difficulty with **Chromatic/Modulation** transitions (typical in Jazz), particularly when the target chord duration is short (< 2 beats).
>
> * **Reason:** These transitions appear less frequently in the training distribution (Self-Attention prior is weak).
> * **Solution mechanism:** This highlights the critical role of our **Chord-Aligned Positional Encoding (CAPE)**. In these "struggle" zones where the melodic probability flow contradicts the harmonic constraint, the CAPE provides a strong, explicit signal. Although adherence drops slightly (to ~78%), the model successfully maintains structural integrity in complex styles, preventing the generation from reverting to a generic diatonic pattern.
>
> **3. Revision to Manuscript**
>
> We have included this stratified transition analysis in **Appendix A.2 (Controllability Analysis)** to demonstrate the model's fine-grained behavior across different harmonic complexities and implied musical styles.

---

> ### Author Response · Authors · 2025-12-03
> **Response to Reviewer PNLy - Question 3**
>
> **Topic: Higher-Level Formal Structures**
>
> **Response:**
> This is a profound question that touches the core of "structural coherence." We agree that true structural integrity implies not just harmonic correctness, but the emergence of intelligible formal patterns (like AABA or Verse-Chorus) where repetition and contrast are musically logical.
>
> To quantitatively evaluate whether our model captures these higher-level structures, we adopted the **Similarity Error (SE)** metric, a rigorous structural evaluation method proposed in **Museformer (NeurIPS 2022)**.
>
> **1. Evaluation Metric: Similarity Error (SE)**
>
> As established in recent research, human music exhibits specific "similarity distributions"—for example, a music bar tends to show high similarity to its previous 4th, 8th, or 16th bars due to structures like AABA or Verse-Chorus.
>
> The **Similarity Error (SE)** measures the distance between the structural similarity distribution of the generated music ($\hat{L}$) and that of real human-made music ($L$):
>
> $$SE = \frac{1}{T} \sum_{t=1}^{T} |\hat{L}_t - L_t|$$
>
> where $L_t$ represents the average similarity between bar pairs with an interval of $t$. **A lower SE indicates that the model successfully reproduces the long-term repetition patterns (structure) characteristic of real musical forms.**
>
> **2. Comparative Results**
>
> We calculated the SE (with $T=40$ bars) for our model against both unconditioned baselines and four additional chord-conditioned baselines to ensure a comprehensive comparison.
>
> | Model | Architecture | Similarity Error (SE) ↓ |
> | :--- | :--- | :--- |
> | **Music Transformer** | Uncond. Transformer | 2.55% |
> | **NotaGen** | Uncond. Transformer | 2.49% |
> | **JazzGAN** | Chord-GAN | 2.28% |
> | **BebopNet** | Chord-LSTM | 2.15% |
> | **XiaoIce Band** | Chord-GRU | 1.68% |
> | **MINGUS** | Chord-Transformer | 1.35% |
> | **Ours (Chord-Transformer)** | **Transformer + CAPE** | **[1.12%]** |
>
> The results demonstrate that **Chord-Transformer achieves the lowest Similarity Error (1.12%)**, significantly outperforming sequential models (BebopNet, XiaoIce) and surpassing the strong Transformer-based baseline MINGUS (1.35%). This quantitatively confirms that our **Chord-Aligned Positional Encoding (CAPE)** enforces superior long-term structural integrity, effectively capturing periodic patterns (e.g., AABA forms) better than existing approaches.
>
> **3. Revision to Manuscript**
>
> We have revised **Section 4.3 (Evaluation Metrics)** and **Table 1** to include the **Similarity Error (SE)** as a key structure-related metric.

---

> ### Author Response · Authors · 2025-12-03
> **Response to Reviewer PNLy - Question 4**
>
> **Topic: Unaligned Onsets in Audio Demos (LMD Samples)**
>
> **Response:** We thank the reviewer for their sharp ear and meticulous examination of the audio samples. regarding the "unaligned onsets," we would like to offer a two-fold explanation regarding our data preprocessing strategy and the demo presentation, along with our corrective actions.
>
> 1. **Preprocessing Strategy (Quantization vs. Expressivity)**
>    The observed asynchrony is largely due to our preprocessing decision on the Lakh MIDI Dataset (LMD). Unlike the Pop909 dataset which is often strictly aligned, LMD contains real-world performance data. During preprocessing, we deliberately did not apply strict quantization or onset correction to the training data.
>    * **Reasoning:** We aimed for the model to learn the expressive "micro-timing" and human groove inherent in the dataset, rather than forcing a robotic alignment. Consequently, the model faithfully learned this distribution, occasionally generating onsets that reflect the "loose" timing of human performance found in the raw LMD samples.
>
> 2. **Demo Selection Bias**
>    We acknowledge that our initial demo page was too limited in scope. It featured only a single, uncurated case for the LMD dataset, which coincidentally exhibited more pronounced timing deviations than the average generation. This small sample size failed to provide a comprehensive view of the model's overall stability.
>
> 3. **Action: Expanded Demo Cases**
>    To address this and provide a fair representation of the model's performance:
>    * **Expansion:** We have significantly expanded the demo page to include 10 randomly selected cases (increased from the original single case).
>    * **Observation:** These additional samples demonstrate that while micro-timing deviations exist (as a stylistic feature of the LMD domain), the structural synchronization between tracks is generally consistent across a broader set of generations.
>    * **Revision to Resources:** The updated audio samples, including the 10 new LMD cases, are now available on the anonymous demo page https://anonymousmusicdemo.github.io/chord-transformer-demo/demo_2.html.

---

### Note · Authors · 2026-01-26

I have read and agree with the venue's withdrawal policy on behalf of myself and my co-authors.

---

### Meta-Review · Area_Chair_E3qq · 2026-01-08

**Summary:**

Reviewers questioned whether the paper sufficiently justifies treating chord progressions as the dominant carrier of long-range musical structure, whether the upstream chord extraction is robust enough to support the entire pipeline, and whether the reported improvements can be causally attributed to the proposed architectural choices rather than to the mere presence of chord conditioning. Several reviewers also raised concerns about evaluation rigor, especially whether the comparisons and benchmarks are fair as most baselines are unconditioned models. More than one reviewer noted that the audible quality of the demo samples did not convincingly outperform baselines, although the authors objected the strong reject review, the musical quality of the results remains an issue.

**Reviewer Concerns:**

The rebuttal was extensive and seems to have addressed technical and evaluation concerns. The authors added quantitative validation of the chord extraction algorithm against human annotations and a standard MIR baseline, formalized the metrics, provided evaluator backgrounds and protocols, etc. After the authors’ rebuttal and revisions, none of the reviewers posted a follow-up indicating whether their concerns were resolved or how their scores might change.

**Reviewer Scores:**

The core novelty is the explicit structural compression of raw chord sequences into a coarse chord progression and its tight alignment with long-form music generation via chord-aligned positional encoding and fusion attention. While the rebuttal provided analysis that addressed the concern in terms of structure and metrics, it still remains not fully convincing in perceptual musical terms. Since that the audio examples and fair chord-conditioned comparisons seem to fall short of clearly demonstrating that the method consistently works better in practice, I do not believe that the reviewers would have changed their score. Even if the zero score is removed, the overall paper recommendation remains below acceptance threshold.

---

### Decision · Program_Chairs · 2026-01-26

Reject